# Adaptive Multi-Round Allocation with Stochastic Arrivals

**Yuqi Pan** [* 1]   **Davin Choo** [* 1]   **Haichuan Wang** [1]   **Milind Tambe** [1]   **Alastair van Heerden** [2 3]   **Cheryl Johnson** [4]

## Abstract

We study a sequential resource allocation problem motivated by adaptive network recruitment, in which a limited budget of identical resources must be allocated over multiple rounds to individuals with stochastic referral capacity. Successful referrals endogenously generate future decision opportunities while allocating additional resources to an individual exhibits diminishing returns. We first show that the single-round allocation problem admits an exact greedy solution based on marginal survival probabilities. In the multi-round setting, the resulting Bellman recursion is intractable due to the stochastic, high-dimensional evolution of the frontier. To address this, we introduce a population-level surrogate value function that depends only on the remaining budget and frontier size. This surrogate enables an exact dynamic program via truncated probability generating functions, yielding a planning algorithm with polynomial complexity in the total budget. We further analyze robustness under model misspecification, proving a multi-round error bound that decomposes into a tight single-round frontier error and a population-level transition error. Finally, we evaluate our method on real-world inspired recruitment scenarios.

## 1. Introduction

Adaptive network-based recruitment arises in a wide range of real-world settings, including respondent-driven sampling (RDS) (Heckathorn, 1997; Goel & Salganik, 2009) and incentivized contact-tracing campaigns (Munzert et al., 2021). In these applications, a decision-maker seeks to grow participation by allocating limited resources, such as referral vouchers, self-test kits, or other incentives, to individuals in an existing network, who may then recruit others. Crucially, recruitment is both *stochastic* and *endogenous*: allocating more resources to an individual can increase the number of recruits they generate, and these recruits in turn create new opportunities for future recruitment.

**Real-world motivation.** To ground our work in a concrete application, we focus on improving RDS as a tool for advancing the 95-95-95 HIV targets proposed by UNAIDS (UNAIDS, 2022).[1] RDS is a network-based chain-referral design for hard-to-reach populations in which a small set of initial participants ("seeds") is given a limited number of referral opportunities, and successful recruits become recruiters in the next wave. This mechanism is widely used in public-health studies (McCreesh et al., 2013; Wylie & Jolly, 2013; Truong et al., 2023; Wang et al., 2025). For example, (Wang et al., 2025) uses RDS to recruit individuals with methamphetamine use disorder in China: seeds receive recruitment cards and incentives, can refer up to five peers, and recruitment proceeds over multiple rounds.

Unlike broadcast advertising, the set of individuals available in future rounds is endogenously determined by current recruitment outcomes. Network-based approaches such as RDS are recommended by WHO (World Health Organization, 2019) because centralized outreach is often ineffective for hard-to-reach or stigmatized populations, whereas peer referral leverages social networks under tight resource constraints. In this and similar incentivized recruitment settings, the overarching objective is to recruit as many individuals as possible, as early as possible, subject to a constrained and irreversible resource budget.

**Our approach.** We formalize this process as a sequential decision problem. At each round, one observes a *frontier* of individuals who are eligible to receive resources. Each individual is characterized by observable covariates (e.g., demographics, location, or network features), which induce a probability distribution over the number of effective referrals the individual can generate. Realizations of

---

[*]Equal contribution  [1]Harvard University  [2]University of Witwatersrand  [3]Wits Health Consortium  [4]World Health Organization. Correspondence to: Yuqi Pan <yuqipan@g.harvard.edu>, Davin Choo <davinchoo@seas.harvard.edu>.

*Proceedings of the 43rd International Conference on Machine Learning*, Seoul, South Korea. PMLR 306, 2026. Copyright 2026 by the author(s).

---

[1]The human immunodeficiency virus (HIV) remains a major global health issue, having caused over 42 million deaths to date (World Health Organization, 2024). In support of the UN Sustainable Development Goal 3.3 (United Nations, n.d.), the 95-95-95 targets aim for 95% of people with HIV to know their status, 95% of those to receive treatment, and 95% of treated individuals to achieve viral suppression.

these referrals are stochastic and *unobserved* at decision time. Given a limited total resource budget, one must decide how many resources to allocate to current frontier, and to each individual. Note that an individual can only make use of resources up to their realized referral capacity, reflecting common occurrences in coupon-based recruitment and incentivized referral programs, where unused coupons or incentives cannot be reassigned once issued (Heckathorn, 1997; Goel & Salganik, 2009; World Health Organization, 2013). Newly recruited individuals then form the frontier for the next round, and the process continues until the budget is exhausted or recruitment dies out.

A central modeling assumption in our work is that an individual's referral behavior can be represented as a count-valued random variable whose distribution depends on observable covariates. This assumption aligns with standard practice in the statistical modeling of count data, where covariates are commonly used to parameterize conditional outcome distributions, rather than producing deterministic point estimates (Hilbe, 2014; Hardin & Hilbe, 2018). Accordingly, we assume that the mapping from covariates to referral behavior can be learned from historical data or domain knowledge: each individual's covariates is abstracted as an *arrival distribution* over $\mathbb{N}$, representing the number of recruits the individual can generate; see Fig. 1. This abstraction allows us to reason about recruitment dynamics without explicitly tracking high-dimensional covariates, while still capturing heterogeneous and uncertain referral behavior.

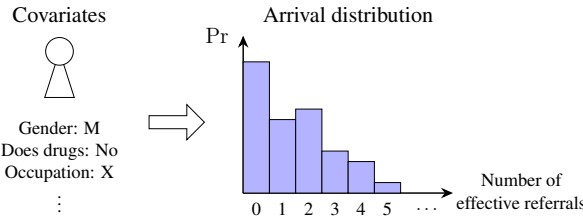

*Figure 1.* The mapping from covariates to an arrival distribution can be learnt from historical data and domain knowledge.

Our setting is related to classical problems such as stochastic knapsack, budgeted bandits, and prophet inequalities; see Section 2. However, it differs in a fundamental way: allocation decisions not only consume budget and yield immediate reward, but also *alter the distribution of future decision opportunities* by changing the size and composition of the frontier. Thus, the decision-maker faces a multi-round stochastic control problem with endogenous state evolution.

**Our contributions.** We model adaptive network recruitment as a budgeted stochastic control problem and show that optimal single-round allocation is greedy. To enable tractable multi-round planning under endogenous arrivals, we introduce a population-level surrogate value function, yielding an efficient and robust policy based on greedy allo-

cation and dynamic programming. In more detail:
**1.** We model multi-round adaptive recruitment as a budgeted stochastic control problem with endogenous arrivals.
**2.** We prove that the single-round allocation problem admits an exact greedy solution for any realized frontier.
**3.** We introduce a tractable population-level surrogate and an exact dynamic program for multi-round planning.
**4.** We provide a multi-round performance guarantee under noisy distributional estimates with interpretable error terms.
**5.** We empirically validated our approach on real-world recruitment networks.

Proofs and experimental details are deferred to the appendix.

## 2. Related work

Our work is related to several strands of literature on budgeted stochastic control, online resource allocation, and sequential decision-making under uncertainty. We highlight the most relevant connections and distinctions below.

**Constrained and budgeted Markov decision processes.** Our problem can be viewed as a finite-horizon, budget-constrained control problem, related to classical finite-horizon MDPs (Puterman, 2014) and constrained MDPs (CMDPs) (Altman, 1999). These formulations typically assume a fixed-dimensional state space and dynamics over a fixed set of entities. Related perspectives include weakly-coupled models, where multiple subproblems interact through shared resource constraints (Meuleau et al., 1998; Dolgov & Durfee, 2005), and budgeted MDPs, which explicitly incorporate resource constraints into the state or objective (Boutilier & Lu, 2016; Carrara et al., 2019).

In contrast, we have a variable-size, distribution-valued state space in our problem setting, akin to branching MDPs (Etessami et al., 2018; 2019), as our allocations stochastically generates the next decision frontier.

**Stochastic knapsack, budgeted bandits, prophet inequalities, and online selection.** The stochastic knapsack problem (Kleywegt & Papastavrou, 1998; Dean et al., 2008) studies sequentially executing jobs with stochastic sizes (revealed upon execution) and known rewards under a fixed capacity constraint, with extensions allowing for reward-size correlations and cancellation (Gupta et al., 2011). Closely related are bandits with knapsacks (Badanidiyuru et al., 2018; Kesselheim & Singla, 2020; Immorlica et al., 2022; Kumar & Kleinberg, 2022), where each action yields stochastic reward while consuming limited resources.

Prophet inequalities (Krengel & Sucheston, 1977; 1978; Samuel-Cahn, 1984) study online selection with sequentially arriving candidates whose values are drawn from known distributions, where decisions are irrevocable, with extension to multi-item selection under feasibility con-

straints (Kleinberg & Weinberg, 2012), allocation of identical goods to sequential agents (Hajiaghayi et al., 2007), and mechanisms robust to prior misspecification (Dütting & Kesselheim, 2019).

A key distinction from our setting is that these models assume a fixed or exogenously evolving set of actions whose availability and distributions are unaffected by past decisions. In contrast, allocation actions in our model affect the distribution of future arrivals, inducing endogenous population dynamics and history-dependent state evolution that fundamentally alter the decision structure.

**Approximate dynamic programming.** Our approach also connects to approximate dynamic programming and value-function approximation, where intractable continuation values are replaced by lower-dimensional surrogates, e.g., via state aggregation and rollout/lookahead methods (Powell, 2007; Bertsekas, 2012; 2025). In contrast, our surrogate is defined at the population level and computed through a structured optimization problem, enabling principled error control while retaining computational tractability.

## 3. Model and Problem Formulation

In this section, we formally define our model.

**Notation.** Scalars are denoted by lowercase letters (e.g., $r, s, n$), vectors by boldface (e.g., $\mathbf{k}$), and random variables by uppercase letters (e.g., $X, N$). Calligraphic letters denote distributions or structured objects (e.g., $\mathcal{D}, \mathcal{P}$). For an integer $n \geq 1$, we write $[n] = \{1, 2, \ldots, n\}$.

**The Frontier** At any round, the set of active individuals is called the *frontier*, and its size is denoted by $n$. Each individual $i \in [n]$ is associated with an arrival distribution $\mathcal{D}_i$ over $\mathbb{N}$, the number of recruits that $i$ can generate. A realized frontier of size $n$ is written as $\mathcal{D}_{1:n} = (\mathcal{D}_1, \ldots, \mathcal{D}_n)$.

**State, actions, and rewards.** At any round, the system is in a state $(r, \mathcal{D}_{1:n})$, where $r \in \mathbb{N}$ denotes the remaining budget and $\mathcal{D}_{1:n}$ is the current frontier of $n$ individuals. An action consists of two components: (i) a choice of round budget $s \in \{0, 1, \ldots, r\}$, and (ii) an allocation $\mathbf{k} = (k_1, \ldots, k_n) \in \mathbb{N}^n$ satisfying $\sum_{i=1}^n k_i \leq s$. We refer to such a vector $\mathbf{k}$ as an $(s, n)$-allocation and it determines the transition to the next state: while an individual *may* refer $X_i \sim \mathcal{D}_i$ recruits, only $\min\{k_i, X_i\}$ will be successfully recruited. Consequently, the aggregate term $N(\mathbf{k}; X_{1:n}) = \sum_{i=1}^n \min\{k_i, X_i\}$ serves a dual role as both the immediate reward and the determinant of the next frontier size. The process terminates when either the budget is exhausted ($r = 0$) or the frontier becomes empty ($n = 0$).

**Population Model** To model heterogeneity and uncertainty, we assume that individual referral distributions are drawn independently from an underlying *population-level distribu-*

*tion* $\mathcal{P}$. Formally, whenever a new individual joins the frontier, their distribution is sampled as $\mathcal{D} \sim \mathcal{P}$. The decision-maker observes the current realizations $\mathcal{D}_{1:n}$, but the distributions of *future* recruits are characterized only by the prior $\mathcal{P}$. This hierarchical structure ($\mathcal{P} \to \mathcal{D} \to X$) allows us to reason about future frontiers without explicitly tracking covariates or identities, while preserving the stochastic structure induced by the arrival process.

Fig. 2 illustrates the schematic timeline of events.

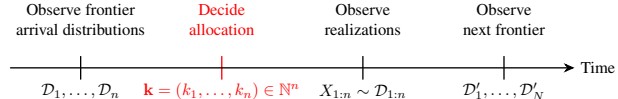

*Figure 2.* Timeline of events: Given arrival distributions $\mathcal{D}_{1:n}$, we make an allocation decision *before* $X_{1:n} \sim \mathcal{D}_{1:n}$ are realized. Note that $N = \sum_{i=1}^n \min\{k_i, X_i\}$ is a random variable, and each arrival distribution $\mathcal{D}_1, \ldots, \mathcal{D}_n, \mathcal{D}'_1, \ldots, \mathcal{D}'_N \sim \mathcal{P}$ are drawn i.i.d. from the population distribution $\mathcal{P}$.

**Budgets dynamics.** We are given a total budget $b \in \mathbb{N}$, which is consumed over multiple rounds. At each round, the remaining budget $r$ constraints the admissible round budget choices $s$ with $0 \leq s \leq r \leq b$. The choice of $s$ determines both the maximum number of resources that can be deployed in the current round and the remaining budget $r - s$ available for future rounds. While the round budget $s$ governs intertemporal trade-offs, the allocation $\mathbf{k}$ determines how resources are distributed among individuals within the current frontier. Importantly, the allocation $\mathbf{k}$ may depend arbitrarily on the realized frontier $\mathcal{D}_{1:n}$.

Time is indexed by $t \geq 1$ when needed, but we suppress the index when focusing on a single round.

**Objective.** A policy $\pi$ maps each state $(r, \mathcal{D}_{1:n})$ to a feasible action $(s, \mathbf{k})$. The goal is to maximize the expected total (discounted) number of recruits generated over the lifetime of the process. We fix a discount factor $\gamma \in (0, 1)$ and evaluate a policy $\pi$ by the reward function $\mathbb{E}_\pi \left[ \sum_{t \geq 1} \gamma^{t-1} \cdot N(\mathbf{k}^{(t)}; X_{1:n^{(t)}}^{(t)}) \right]$, where $n^{(t)}$ is the number of distributions at round $t$ and $\mathbf{k}^{(t)}$ is our decided allocation. Note that $N(\mathbf{k}^{(t)}; X_{1:n^{(t)}}^{(t)})$ is a random variable that depends on both our allocation and random realizations.

## 4. Single-Round Allocation

We begin by isolating the *single-round allocation problem*, which captures a core combinatorial structure underlying our model. In the multi-round setting, allocation decisions affect future frontiers through stochastic recruitment. However, conditional on a realized frontier and a fixed budget for the current round, the allocation problem is static and admits a clean characterization. As such, throughout this section, we fix an arbitrary round and suppress the time index.

**Single-round optimization problem.** Given a frontier of $n$ individuals with arrival distributions $\mathcal{D}_{1:n} = (\mathcal{D}_1, \ldots, \mathcal{D}_n)$ and a round budget $s \in \mathbb{N}$, the decision-maker chooses an $(s, n)$-allocation $\mathbf{k} \in \mathbb{N}^n$ with the goal of maximizing the expected number of recruits generated in the next round:

$$v_s(\mathcal{D}_{1:n}) = \max_{\mathbf{k}:\sum_i k_i \leq s} \mathbb{E}_{X_{1:n} \sim \mathcal{D}_{1:n}} \left[ \sum_{i=1}^{n} \min\{k_i, X_i\} \right] \quad (1)$$

Unless otherwise stated, expectations throughout this section are taken over the realizations $X_{1:n}$, conditional on the observed arrival distributions $\mathcal{D}_{1:n}$.

The single-round objective in Eq. (1) involves a non-linear, stochastic reward whose marginal value depends on the tail of each referral distribution. Classical stochastic knapsack results do not directly apply as they consider sequential item selection with intermediate feedback, whereas our setting requires simultaneous allocation of a shared budget before any realizations are observed. Despite this, the objective exhibits a natural diminishing-returns structure: allocating additional resources to the same individual becomes progressively less valuable. We make this structure explicit via survival probabilities, which decompose the expected reward into non-increasing marginal contributions.

For a distribution $\mathcal{D}$ over $\mathbb{N}$ and a level $\ell \in \mathbb{N}$, we define the survival (tail) probability as $p_{\mathcal{D}}(\ell) = \Pr_{X \sim \mathcal{D}}(X \geq \ell)$. For a realized frontier $\mathcal{D}_{1:n}$, we write $p_i(\ell) = p_{\mathcal{D}_i}(\ell)$.

**Proposition 4.1** (Marginal decomposition). *For any allocation $\mathbf{k} \in \mathbb{N}^n$, we have*

$$\mathbb{E}\left[ \sum_{i=1}^{n} \min\{k_i, X_i\} \right] = \sum_{i=1}^{n} \sum_{\ell=1}^{k_i} p_i(\ell)$$

This observation of discrete concave (diminishing-returns) structure naturally motivates a greedy allocation policy.

**Greedy allocation policy.** Given a frontier $\mathcal{D}_{1:n}$ and round budget $s$, initialize $k_1 = \cdots = k_n = 0$. For each $\ell = 1, \ldots, s$, allocate the next unit of budget to any individual $i^\star \in \arg\max_{i \in [n]} p_i(k_i + 1)$, and increment $k_{i^\star}$ by one. We denote the resulting allocation by $\mathbf{k}^{\text{greedy}}$, and write $N_s^g = N(\mathbf{k}^{\text{greedy}}; X_{1:n})$ for the corresponding number of recruits generated in the round.

**Theorem 4.2** (Optimality of greedy allocation). *For any realized frontier $\mathcal{D}_{1:n}$ and any round budget $s$, the greedy allocation $\mathbf{k}^{\text{greedy}}$ maximizes the single-round objective of $v_s(\mathcal{D}_{1:n}) = \mathbb{E}_{X_{1:n} \sim \mathcal{D}_{1:n}}[N_s^g]$.*

Theorem 4.2 shows that the intra-round allocation problem admits a deterministic optimal solution depending only on the observed arrival distributions and the round budget. Consequently, we can restrict our attention to greedy single-round allocation and focus solely on choosing the

round budget $s$. This separation enables a clean Bellman formulation of the multi-round problem, developed next.

# 5. Multi-Round Planning

We now return to the full multi-round problem. In contrast to the single-round setting, allocation decisions in one round affect the distribution of future frontiers through stochastic recruitment. We formalize this dependence via a dynamic programming formulation and identify the main computational obstacle that motivates our surrogate approach.

**Optimal value function.** Recall our model from Section 3. Let $V_{\mathcal{P}}(r, \mathcal{D}_{1:n})$ denote the optimal expected total reward achievable starting from state $(r, \mathcal{D}_{1:n})$, where $r$ is the remaining budget and $\mathcal{D}_{1:n}$ is the current frontier. Then, the optimal value satisfies the Bellman equation

$$
\begin{aligned}
V_{\mathcal{P}}(r, \mathcal{D}_{1:n}) = \max_{0 \leq s \leq r} &\max_{\mathbf{k}:\sum_i k_i \leq s} \mathbb{E}_{X_{1:n} \sim \mathcal{D}_{1:n}} \Bigg[ N(\mathbf{k}; X_{1:n}) \\
&+ \gamma \cdot \mathbb{E}_{\mathcal{D}'_{1:N(\mathbf{k})}|N(\mathbf{k})} \Big[ V_{\mathcal{P}}(r - s, \mathcal{D}'_{1:N(\mathbf{k})}) \Big] \Bigg] \quad (2)
\end{aligned}
$$

where, conditioned on $N(\mathbf{k}) = m$, the next frontier satisfies $\mathcal{D}'_{1:m} \sim \mathcal{P}^{\otimes m}$. Here, the action consists of both a round-budget decision $s$ and an allocation $\mathbf{k}$ across individuals.

**Restricting to greedy within-round allocation.** For any fixed round budget $s$, Theorem 4.2 shows that the greedy allocation $\mathbf{k}^{\text{greedy}}$ constructed from the realized frontier $\mathcal{D}_{1:n}$ maximizes the immediate expected reward, with $N_s^g$ denoting the resulting number of recruits. We therefore restrict attention to policies that allocate greedily within each round.[2] Under this within-round rule, the achievable value satisfies the recursion Eq. (3). The per-round action then reduces to the single scalar choice of $s$, with greedy allocation applied for any fixed budget $s$.

$$
\begin{aligned}
V_{\mathcal{P}}(r, \mathcal{D}_{1:n}) = \max_{0 \leq s \leq r} \mathbb{E}_{X_{1:n} \sim \mathcal{D}_{1:n}} \Bigg[ N_s^g \\
+ \gamma \cdot \mathbb{E}_{\mathcal{D}'_{1:N_s^g}} \Big[ V_{\mathcal{P}}(r - s, \mathcal{D}'_{1:N_s^g}) \Big] \Bigg] \quad (3)
\end{aligned}
$$

where, conditioned on $N_s^g = m$, the next frontier satisfies $\mathcal{D}'_{1:m} \sim \mathcal{P}^{\otimes m}$. That is, the action space now only consists of the round-budget decision $s$ as we can apply the greedy allocation policy for any fixed budget $s$ due to Theorem 4.2.

---

[2]Strictly speaking, this is a restriction to greedy within-round allocation rather than an exact reduction of Eq. (2). Greedy maximizes the immediate expected reward $\mathbb{E}[N(\mathbf{k}; X_{1:n})]$, whereas the inner maximization in Eq. (2) also involves the continuation term, which depends on the full distribution of the next frontier size rather than only its mean.

## 6. A Tractable Population-Level Surrogate

Unfortunately, despite the simplification afforded by greedy single-round optimality, Eq. (3) remains intractable to compute or represent exactly. There are two fundamental obstacles. First, the size of the next frontier $N_s^g$ is a random variable depending on the realized arrivals $X_{1:n}$. As a result, the dimension of the next state $\mathcal{D}'_{1:N_s^g}$ is itself random, preventing a fixed-dimensional state representation. Second, even conditioning on a fixed value $N_s^g = m$, the continuation term requires evaluating $\mathbb{E}_{\mathcal{D}_{1:m} \sim \mathcal{P}^{\otimes m}} \left[ V_{\mathcal{P}}(r - s, \mathcal{D}_{1:m}) \right]$, which cannot be computed or stored compactly without strong structural assumptions on $\mathcal{P}$. These difficulties persist even when the population distribution $\mathcal{P}$ is fully known.

To overcome this difficulty, we introduce a population-level surrogate value function that abstracts away the unobserved composition of future frontiers while retaining exact expectations under the population model.

### 6.1. Single-round population-level surrogate

Several natural alternatives (such as planning based only on expected frontier size or approximating continuation values via Monte Carlo simulation) prove insufficient, as they either break the Bellman structure or incur uncontrolled error accumulation due to the large state space. Instead, our population-level surrogate proposed in this section is designed to preserve decision-relevant information while admitting a tractably computable Bellman recursion.

Newly recruited individuals have arrival distributions drawn i.i.d. from the population distribution $\mathcal{P}$, and are therefore ex ante indistinguishable at decision time. Define the population survival probabilities $\bar{p}(\ell) = \Pr_{\substack{\mathcal{D} \sim \mathcal{P} \\ X \sim \mathcal{D}}}(X \geq \ell)$ and $g(k) = \sum_{\ell=1}^{k} \bar{p}(\ell)$, with $g(0) = 0$. That is, allocating $k$ units of budget to a single individual drawn from $\mathcal{P}$ yields expected immediate reward $g(k)$. For a size $n$ frontier and round budget $s$, the population-level single-round problem

$$\max_{\substack{k_1, \ldots, k_n \in \mathbb{N} \\ \sum_{i=1}^{n} k_i = s}} \sum_{i=1}^{n} g(k_i) \quad (4)$$

is a separable concave integer optimization problem.

Under a population model where individuals are ex ante identical, one might conjecture that an even allocation is optimal by symmetry. However, symmetry alone does not directly imply optimality in our setting since allocating an additional unit affects both the immediate reward and future uncertainty through a truncated, nonlinear response. Instead, we exploit the discrete concavity of Eq. (4) to establish that an even allocation is indeed optimal at the population level.

**Even allocation.** Let $a = \lfloor s/n \rfloor$ and $c = s - a \cdot n$, so that $0 \leq c < n$ and $s = a \cdot n + c$. An *even allocation* assigns

$k_i \in \{a, a+1\}$ for all $i \in [n]$, with exactly $c$ individuals receiving $a + 1$ units and the other $n - c$ receiving $a$ units.

**Proposition 6.1** (Even allocation is optimal under $\mathcal{P}$). *Fix $n, s \in \mathbb{N}$, and define $a = \lfloor s/n \rfloor$ and $c = s - a \cdot n$. Any optimal solution to Eq. (4) satisfies $k_i \in \{a, a+1\}$ for all $i \in [n]$, with exactly $c$ indices satisfying $k_i = a + 1$.*

Proposition 6.1 formalizes the intuition that, under exchangeability and diminishing returns, spreading budget as evenly as possible is optimal. The resulting population-level single-round value is then

$$\bar{v}_s(n) = n \sum_{\ell=1}^{a} \bar{p}(\ell) + c \cdot \bar{p}(a+1)$$

Note that the population-level term $\bar{v}_s(n)$ depends only on the frontier size while the single-round objective term $v_s(\mathcal{D}_{1:n})$ in Eq. (1) depends on the realized distributions.

### 6.2. Population-level surrogate value function

We now define a surrogate value function that depends only on the remaining budget and the frontier size.

Let $U_{\mathcal{P}}(r, n)$ denote the optimal expected total number of recruits obtainable given remaining budget $r$ and frontier size $n$, where the $n$ individuals' arrival distributions are drawn independently from $\mathcal{P}$ but are not observed individually.

The surrogate value function is defined recursively as

$$U_{\mathcal{P}}(r, n) = \max_{0 \leq s \leq r} \mathbb{E}_{\substack{\mathcal{D}_{1:n} \sim \mathcal{P}^{\otimes n} \\ X_{1:n} \sim \mathcal{D}_{1:n}}} \left[ N_s^e + \gamma \cdot U_{\mathcal{P}}(r - s, N_s^e) \right] \quad (5)$$

where $N_s^e = N(\mathbf{k}^{\text{even}}; X_{1:n})$ denotes the number of recruits generated by the even allocation with parameters $(a, c)$ defined above. Note that the boundary conditions are $U_{\mathcal{P}}(0, n) = 0$ for all $n$ and $U_{\mathcal{P}}(r, 0) = 0$ for all $r$.

### 6.3. Bellman operator for the population-level surrogate

Given a realized frontier $\mathcal{D}_{1:n}$, the greedy allocation $\mathbf{k}^{\text{greedy}}$ is single-round optimal (Theorem 4.2). Replacing the true continuation value in Eq. (3) by the population-level surrogate yields the modified Bellman operator

$$\widetilde{V}_{\mathcal{P}; U_{\mathcal{P}}}(r, \mathcal{D}_{1:n}) = \max_{0 \leq s \leq r} \mathbb{E}_{X_{1:n} \sim \mathcal{D}_{1:n}} \left[ N_s^g + \gamma \cdot U_{\mathcal{P}}(r - s, N_s^g) \right]$$

$$(6)$$

where $N_s^g = N(\mathbf{k}^{\text{greedy}}; X_{1:n})$. This operator preserves optimality of the current-round decision while planning over future uncertainty through the population-level dynamic program $U_{\mathcal{P}}$. It therefore yields a tractable approximation to the original Bellman equation that can be computed exactly.

### 6.4. Computing the population-level Bellman equation

To evaluate Eq. (5), it suffices to characterize the distribution of the next frontier size under the even allocation. To do this, we use the notion of probability generating functions (PGFs). In Appendix A, we describe how PGF properties allow us to yield the exact transition probabilities required to compute $\mathbb{E}[U_{\mathcal{P}}(r - s, N_s^e)]$. Consequently, the population-level Bellman equation can be solved via dynamic programming using truncated polynomial arithmetic. Precomputing the surrogate table is a one-time offline cost, once available, online execution only requires solving a single-round greedy allocation together with evaluating the surrogate table.

**Theorem 6.2** (Computational complexity of the surrogate). *For a total budget of $b$, the population-level value function $U_{\mathcal{P}}(r, n)$ can be computed exactly for all $0 \leq n \leq r \leq b$ using $\mathcal{O}(b^2)$ space and $\mathcal{O}(b^5 \log b)$ time.*

## 7. Robustness Analysis

Thus far, we have assumed idealized access to the individual referral distributions $\mathcal{D}_{1:n}$ and the population distribution $\mathcal{P}$. In practice, both must be estimated from data and are therefore noisy. In this section, we analyze the performance of our algorithm under such imperfect information.

At a state $(r, \mathcal{D}_{1:n})$, the algorithm observes only estimates $\widehat{\mathcal{D}}1:n$ and $\widehat{\mathcal{P}}$, and selects a round budget

$$s^{\mathrm{us}} \in \arg\max_{0 \leq s \leq r} \mathbb{E}_{X_{1:n} \sim \widehat{\mathcal{D}}_{1:n}} \left[ \widehat{N}_s^g + \gamma \cdot U_{\widehat{\mathcal{P}}}(r - s, \widehat{N}_s^g) \right] \quad (7)$$

followed by greedy within-round allocation. Our goal is to bound the resulting suboptimality relative to the best greedy within-round policy $\pi^*$ with access to $(\mathcal{D}_{1:n}, \mathcal{P})$:

$$V_{\mathcal{P}}^{\pi^*}(r, \mathcal{D}_{1:n}) - V_{\mathcal{P}}^{\pi^{\mathrm{our}}}(r, \mathcal{D}_{1:n}) \quad (8)$$

Our analysis preserves a *tight* single-round regret guarantee of the greedy allocation while extending it to the multi-round setting via a stability analysis of the Bellman recursion.

### 7.1. Single-round error under noisy distributions

For a fixed round budget $s \in \mathbb{N}$ and frontier $\mathcal{D}_{1:n}$, let $v_s(\mathcal{D}_{1:n})$ denote the optimal expected single-round reward under the true distributions, and let $v_s(\widehat{\mathcal{D}}_{1:n})$ denote the expected reward obtained by running the greedy allocation using the estimated distributions $\widehat{\mathcal{D}}_{1:n}$.

**Proposition 7.1** (Tight single-round error bound). *Fix a round budget $s \in \mathbb{N}$. For any frontier distributions $\mathcal{D}_{1:n}$ and noisy estimates $\widehat{\mathcal{D}}_{1:n}$,*

$$v_s(\mathcal{D}_{1:n}) - v_s(\widehat{\mathcal{D}}_{1:n}) \leq \sum_{i=1}^n \sum_{\ell=1}^s \left| p_{\mathcal{D}_i}(\ell) - p_{\widehat{\mathcal{D}}_i}(\ell) \right|$$

*Moreover, this bound is tight in the worst case.*

Proposition 7.1 shows that the immediate loss incurred by using noisy estimates is controlled exactly by discrepancies in survival probabilities with $\ell \leq s$, and that no improvement is possible in general without additional assumptions.

### 7.2. Multi-round error decomposition

Bounding estimation error in our setting is challenging for two reasons. First, errors in $\widehat{\mathcal{D}}_{1:n}$ affect both the greedy allocation and the round-budget decision. Second, these errors propagate forward through endogenous recruitment, influencing the entire future trajectory. By carefully separating frontier-level, surrogate, and population-model errors, the following result provides a meaningful multi-round bound on Eq. (8) that preserves tight single-round guarantees.

**Theorem 7.2** (Multi-round error decomposition). *Fix a state $(r, \mathcal{D}_{1:n})$, population $\mathcal{P}$, and discount factor $\gamma \in (0, 1)$. Let $\pi^*$ be optimal for $V_{\mathcal{P}}$, and let $\pi^{\mathrm{our}}$ be a policy whose first-step decision at $(r, \mathcal{D}_{1:n})$ is a round budget $s^{\mathrm{us}}$ defined in Eq. (7), after which it uses the greedy allocation at that budget. Then,*

$$V_{\mathcal{P}}^{\pi^*}(r, \mathcal{D}_{1:n}) - V_{\mathcal{P}}^{\pi^{\mathrm{our}}}(r, \mathcal{D}_{1:n})$$

$$\leq 2(1 + \gamma)r \cdot \sum_{i=1}^n \|\mathcal{D}_i - \widehat{\mathcal{D}}_i\|_{\mathrm{TV}}$$

$$+ c_{r,\gamma} \cdot \|\mathcal{P} - \widehat{\mathcal{P}}\|_{\mathrm{TV}} + c_{r,\gamma} r \cdot \mathbb{E}_{\mathcal{D} \sim \mathcal{P}} \|\mathcal{D} - \bar{\mathcal{D}}\|_{\mathrm{TV}}$$

*where $c_{r,\gamma} = \frac{2\gamma r}{1-\gamma}$ depends only on $r$ and $\gamma$, $\|\cdot\|_{\mathrm{TV}}$ is the total variation distance between two distributions, and $\bar{\mathcal{D}} = \mathbb{E}_{\mathcal{D} \sim \mathcal{P}}[\mathcal{D}]$ is the mean distribution.*

**Interpretation of the bound.** Theorem 7.2 decomposes the suboptimality of the surrogate-based policy into three sources of error. The first $(1 + \gamma)r \cdot \sum_{i=1}^n \|\mathcal{D}_i - \widehat{\mathcal{D}}_i\|_{\mathrm{TV}}$ term captures the immediate loss from noisy frontier-level estimates. It preserves the tight single-round dependence on distributional error while accumulating linearly with the remaining budget $r$, reflecting that early misallocations can affect all subsequent rounds. The second $c_{r,\gamma} \cdot \|\mathcal{P} - \widehat{\mathcal{P}}\|_{\mathrm{TV}}$ term quantifies sensitivity to population-model misspecification and reflects errors in predicting how many new individuals enter the frontier. Its dependence on $r$ and $\gamma$ through $c_{r,\gamma}$ arises from the effective planning horizon. The third $c_{r,\gamma} r \cdot \mathbb{E}_{\mathcal{D} \sim \mathcal{P}} \|\mathcal{D} - \bar{\mathcal{D}}\|_{\mathrm{TV}}$ term captures intrinsic approximation error from replacing the full distribution-valued state by a population-level surrogate. This term vanishes when the population is homogeneous and grows with heterogeneity.

**On the dependence on $r$.** The linear dependence on the remaining budget is unavoidable in the worst case as an error incurred early can influence all future decisions. Such horizon-dependent sensitivity is standard in finite-horizon and discounted MDPs, where value-function perturbations scale linearly with the horizon or as $1/(1 - \gamma)$ in the worst

case (Altman, 1999; Munos & Szepesvári, 2008; Puterman, 2014). While the bound is conservative, experiments show that the surrogate-based policy performs well in practice.

# 8. Experiments

We evaluate the relevance of our approach on a real-world inspired recruitment setting derived from a de-identified, public-use dataset released by ICPSR (Morris & Rothenberg, 2011). The dataset was originally collected to study how social and partnership networks influence the transmission of sexually transmitted and blood-borne infections, and contains social contact networks annotated with demographic covariates (e.g., gender, housing status, and employment status), as well as reported disease status for multiple infections. This setting closely matches the motivating assumptions of our model: recruitment proceeds through social ties, referral capacity is heterogeneous and uncertain, and incentives (e.g., coupons or testing kits) are limited and irreversible. Preprocessing details are provided in Appendix D.

**Empirical population with simulated referrals.** We first construct an empirical population model by grouping individuals with similar covariates using a decision-tree–based partitioning. Within each group, we estimate an empirical distribution over observed degrees, yielding a collection of referral distributions and an induced population distribution $\mathcal{P}$. At execution time, referral counts are sampled stochastically from these learned distributions. This experiment incorporates realistic heterogeneity and covariate structure, under the assumption that the population distribution $\mathcal{P}$ is known.

**Empirical population with realized referrals.** Using the same empirical population model constructed above, we evaluate policies directly on the actual observed network structure. Allocating $k_i$ units of budget to an individual corresponds to selecting up to $k_i$ unrecruited neighbors in the network. This removes distributional assumptions at execution time and evaluates policies under fully realized recruitment dynamics, closely reflecting how RDS operates in practice. Here, the population model $\widehat{\mathcal{P}}$ used by the algorithm may be misspecified, allowing us to test robustness under realistic modeling error.

**Policies compared** Prior work offers few alternatives for our multi-round setting. The most common strategy in practice is a *constant-allocation* policy that assigns each individual a fixed number $k$ of resources (Heckathorn, 1997; Goel & Salganik, 2009). Across all policies, we report the *accumulated discounted reward* $\sum_{t \geq 1} \gamma^{t-1} R_t$, capturing the trade-off between recruiting many individuals and recruiting them early. Results are shown for multiple discount factors $\gamma \in (0, 1)$.

*Constant policies*: Operational guidelines typically use small values such as $k = 2$ or $k = 3$, e.g., see Annex 6 of (World Health Organization, 2013), but we evaluate a broader range $k \in \{2, 3, 5, 10\}$ to span conservative to aggressive regimes. We denote these by `Const(k)`.

*Greedy policies*: Recall from Theorem 4.2 that greedy allocation is optimal for a fixed single-round budget. We therefore pair greedy within-round allocation with different round-budget schedules to isolate the effect of budget scheduling, while keeping the within-round rule fixed to the single-round–optimal greedy assignment. For $\alpha \in \{0.1, 0.2, 0.5, 1.0\}$ and total budget $b$, we either apply the single-round greedy allocation with fixed budget $\alpha b$ per round, or allocate a fraction $\alpha$ of the remaining budget each round. We denote these by `Greedy(α)` and `GreedyRemainder(α)` respectively.

These baselines allow us to benchmark our *surrogate-based policy* $\pi^{\text{our}}$ that adaptively combines greedy within-round allocation with population-level planning, demonstrating the benefit of multi-round planning beyond myopic allocation.

## 8.1. Results and analysis

Here, we present representative plots of our experimental results on HIV network with maximum budget $b = 200$, discount factors $\gamma \in \{0.5, 0.7, 0.9\}$ and intial frontier size $n \in \{5, 10, 15\}$, including both on simulated distributions (see Figure 3) and actual network (see Figure 4). We provide additional experimental results for different disease networks in Appendix D. The qualitative trends and conclusions discussed here remain robust across all those settings.

For the overall quantative trend, we observe that both the constant and greedy baselines exhibit a concave trend with respect to $k$ or $\alpha$. This confirms that choosing constants that are either too small or too large leads to suboptimal performance. For most settings, our policy $\pi^{\text{our}}$ outperforms even the best-tuned constant baselines.

A key takeaway from this comparison is that our approach effectively avoids the need to select a single "best" constant $k$, which can only be identified in hindsight for each instance, and instead replaces this brittle choice with a data-driven policy that leverages historical information to estimate arrival distributions and adapt allocation decisions accordingly.

# 9. Conclusion and discussion

We studied a multi-round, budgeted resource allocation problem motivated by adaptive network recruitment, where allocation decisions exhibit diminishing returns and endogenously shape future recruitment opportunities. Our analysis revealed a clean structural decomposition: within each round, greedy allocation is single-round optimal, and we

Real-world inspired experiments on simulated distributions from ICPSR HIV disease network

30 runs with maximum budget $b = 200$

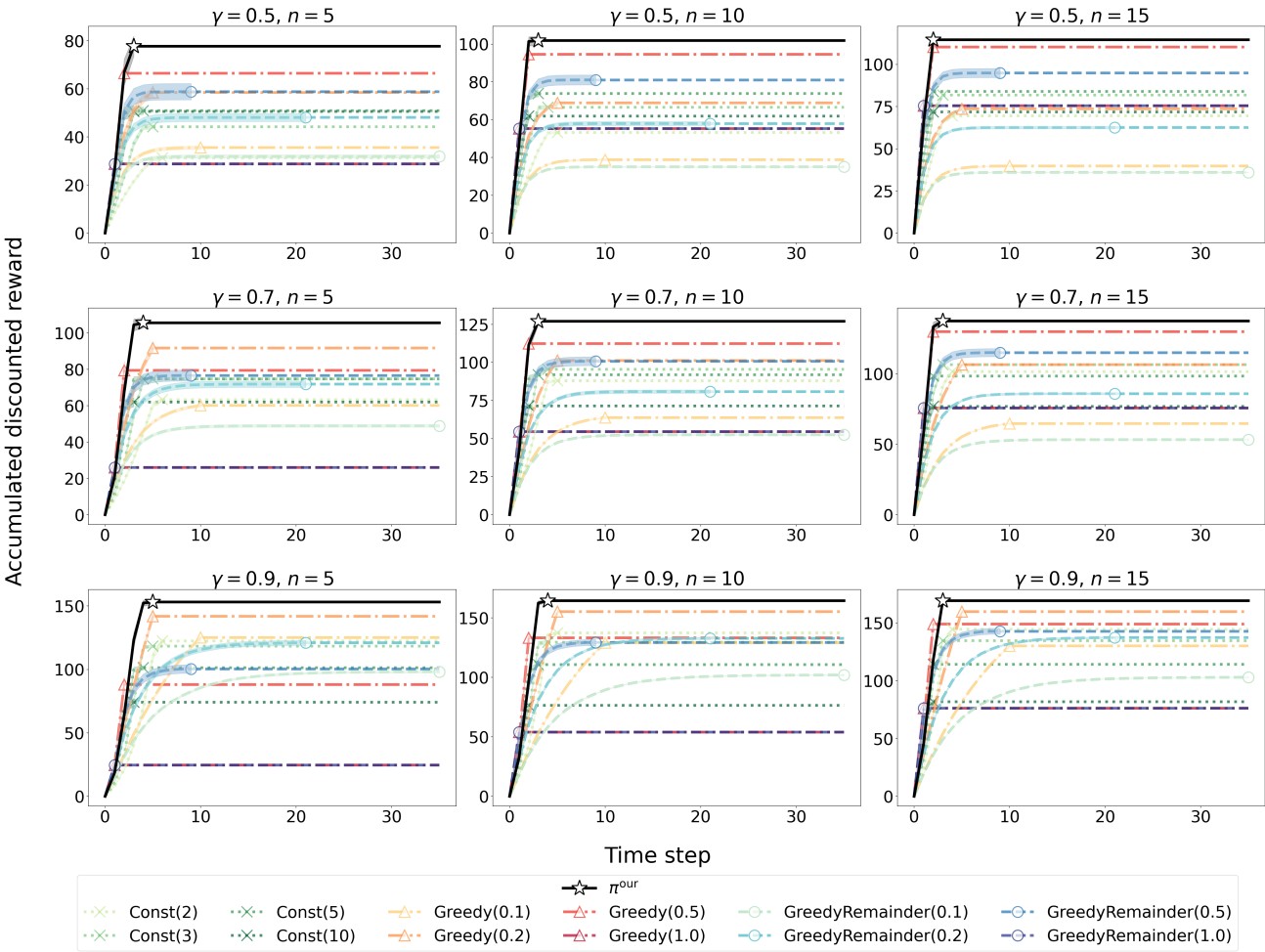

*Figure 3.* Experimental plots for simulated referrals from ICPSR HIV network; see Appendix D for other diseases. We compare our proposed policy $\pi^{\text{our}}$ (solid black line) against the constant- and greedy-allocation baselines described in Section 8 for discount factors $\gamma \in \{0.5, 0.7, 0.9\}$ and initial frontier size $n \in \{5, 10, 15\}$. Each configuration is evaluated over 30 independent runs and we report mean accumulated discounted reward with standard error bands. Markers indicate the termination of the recruitment process, either due to budget exhaustion or an empty frontier.

plan across rounds by selecting round budgets under this greedy within-round policy. To address the intractability of exact planning, we introduced a population-level surrogate value function that captures future uncertainty while remaining efficiently computable. This surrogate enables principled planning with polynomial complexity and admits performance guarantees that decompose suboptimality into interpretable frontier-level and population-level error terms. Experiments on real-world networks demonstrate that the resulting policy performs well in practice and degrades gracefully under model misspecification.

Several directions for future work are promising. One avenue is to explore richer forms of distributional information or alternative surrogate constructions that may yield

sharper approximation guarantees, potentially drawing connections to prophet inequalities and related online selection frameworks. Another is to extend our framework to handle additional real-world operational constraints and possibly strategic participant behavior (e.g., individuals misrepresenting their covariates to alter their perceived $\mathcal{D}$). Finally, our proposed method does not *always* perform well across all experimental settings. For instance, in Appendix D, we see that greedy-based methods outperform us in Chlamydia and Gonorrhea networks for discount factor $\gamma = 0.9$. This suggests that in cases of significant model misspecification and increased requirements for long-term planning (high $\gamma$), our population-based surrogate estimation is suboptimal. Thus, it is a natural future direction is to investigate better

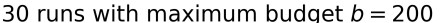

Real-world inspired experiments on actual ICPSR HIV disease network

30 runs with maximum budget $b = 200$

*Figure 4.* Experimental plots for realized referrals from ICPSR HIV network; see Appendix D for other diseases. We compare our proposed policy $\pi^{\text{our}}$ (solid black line) against the constant- and greedy-allocation baselines described in Section 8 for discount factors $\gamma \in \{0.5, 0.7, 0.9\}$ and initial frontier size $n \in \{5, 10, 15\}$. Each configuration is evaluated over 30 independent runs and we report mean accumulated discounted reward with standard error bands. Markers indicate the termination of the recruitment process, either due to budget exhaustion or an empty frontier.

efficiently computable surrogates.

## Acknowledgements

This work was supported by ONR MURI N00014-24-1-2742. The findings and conclusions in this report are those of the authors and do not necessarily represent the official position of the WHO.

## Impact Statement

This paper advances the theory and practice of sequential decision-making under uncertainty by developing tractable planning methods for multi-round, budget-constrained allocation with endogenous dynamics. The primary societal relevance lies in applications such as adaptive network recruitment, RDS, and incentivized public health outreach, where efficient use of limited resources may improve coverage and timeliness of interventions. We note that prioritization based on estimated recruitment potential could interact with existing structural inequalities if underlying data reflect uneven network access or participation. Our framework is intended as a decision-support tool rather than an automated policy prescription, and its deployment should be accompanied by appropriate domain oversight and safeguards. Beyond these considerations, we do not foresee additional ethical concerns specific to this work beyond those already well understood in machine learning and stochastic decision-making.

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

# A. Probability generating functions (PGFs)

In this section, we elaborate on our use of probability generating functions (PGFs) for computing the population-level Bellman equation, as discussed in Section 6.4.

To compute the distribution of the next frontier size under a fixed allocation, we use probability generating functions (PGFs). For a nonnegative integer-valued random variable $Y$, its PGF is defined as $G_Y(z) = \mathbb{E}[z^Y]$ where $z \in \mathbb{R}$. PGFs compactly encode the full distribution of $Y$: the probability mass function can be recovered via $\Pr(Y = m) = [z^m]G_Y(z)$, where $[z^m](\cdot)$ denotes the coefficient of $z^m$.

Two properties of PGFs are particularly useful for us. Firstly, if $Y_1, \ldots, Y_k$ are independent, then $G_{Y_1 + \cdots + Y_k}(z) = \prod_{i=1}^{k} G_{Y_i}(z)$. Secondly, truncating a random variable at level $s$ corresponds to truncating its PGF at degree $s$, with all remaining probability mass accumulated at the $z^s$ term.

Now, for a distribution $\mathcal{D}$ and truncation level $s$, let us define the truncated PGF $G_{\mathcal{D},s}(z)$ and its population-averaged variant $\bar{G}_s(z)$ as follows:

$$G_{\mathcal{D},s}(z) = \mathbb{E}[z^{\min\{X,s\}}]$$
$$= \sum_{\ell=0}^{s-1} \Pr_{X \sim \mathcal{D}}(X = \ell)\, z^\ell + \Pr_{X \sim \mathcal{D}}(X \geq s)\, z^s$$
$$\bar{G}_s(z) = \mathbb{E}_{\mathcal{D} \sim \mathcal{P}}[G_{\mathcal{D},s}(z)]$$

Under the even allocation with parameters $(a, c)$, the next frontier size $N_s^e$ satisfies

$$\mathbb{E}[z^{N_s^e}] = \bar{G}_a(z)^{n-c} \cdot \bar{G}_{a+1}(z)^c$$

and hence

$$\Pr(N_s^e = m) = [z^m]\left(\bar{G}_a(z)^{n-c} \cdot \bar{G}_{a+1}(z)^c\right)$$

These coefficients give the exact transition probabilities required to compute $\mathbb{E}[U_{\mathcal{P}}(r - s, N_s^e)]$. As a result, the population-level Bellman equation can be solved by dynamic programming using truncated polynomial arithmetic.

# B. Multi-round decomposition

In this section, we show how to prove Theorem 7.2, restated below for convenience, through a series of lemmas. The full proofs of these lemmas are deferred to Appendix C.

**Theorem B.1** (Theorem 7.2 restated)**.** *Fix a state $(r, \mathcal{D}_{1:n})$, population $\mathcal{P}$, and discount factor $\gamma \in (0, 1)$. Let $\pi^*$ be optimal for $V_{\mathcal{P}}$, and let $\pi^{\mathrm{our}}$ be a policy whose first-step decision at $(r, \mathcal{D}_{1:n})$ is a round budget $s^{\mathrm{us}}$, after which it uses the greedy allocation at that budget. Then,*

$$V_{\mathcal{P}}^{\pi^*}(r, \mathcal{D}_{1:n}) - V_{\mathcal{P}}^{\pi^{\mathrm{our}}}(r, \mathcal{D}_{1:n}) \leq 2(1+\gamma)r \cdot \sum_{i=1}^{n} \|\mathcal{D}_i - \widehat{\mathcal{D}}_i\|_{\mathrm{TV}} + c_{r,\gamma} \cdot \|\mathcal{P} - \widehat{\mathcal{P}}\|_{\mathrm{TV}} + c_{r,\gamma} r \cdot \mathbb{E}_{\mathcal{D} \sim \mathcal{P}} \|\mathcal{D} - \bar{\mathcal{D}}\|_{\mathrm{TV}}$$

*where $c_{r,\gamma} = \frac{2\gamma r}{1-\gamma}$ depends only on $r$ and $\gamma$, $\|\cdot\|_{\mathrm{TV}}$ is the total variation distance between two distributions, and $\bar{\mathcal{D}} = \mathbb{E}_{\mathcal{D} \sim \mathcal{P}}[\mathcal{D}]$ is the mean distribution.*

Our first step is to reduce the value-function gap between policies to a difference in *one-step Bellman objectives*.

Recall the greedy Bellman equation from Eq. (3). For fixed $(r, \mathcal{D}_{1:n})$ and $s \leq r$, define the corresponding one-step Bellman objective

$$J_{\mathcal{P}}(s; r, \mathcal{D}_{1:n}) = \mathbb{E}_{X_{1:n} \sim \mathcal{D}_{1:n}}\left[N_s^g + \gamma \cdot \mathbb{E}_{\mathcal{D}'_{1:N_s^g}}\left[V_{\mathcal{P}}(r - s, \mathcal{D}'_{1:N_s^g})\right]\right] \tag{9}$$

That is, $V_{\mathcal{P}}(r, \mathcal{D}_{1:n}) = \max_{0 \leq s \leq r} J_{\mathcal{P}}(s; r, \mathcal{D}_{1:n})$. Let us also define $\widehat{J}(s; r, \widehat{\mathcal{D}}_{1:n})$ as the corresponding term when executing our policy $\pi^{\mathrm{our}}$ on noisy distributional estimates.

$$\widehat{J}_{\widehat{\mathcal{P}}}(s; r, \widehat{\mathcal{D}}_{1:n}) = \mathbb{E}_{\widehat{X}_{1:n} \sim \widehat{\mathcal{D}}_{1:n}}\left[\widehat{N}_s^g + \gamma \cdot U_{\widehat{\mathcal{P}}}(r - s, \widehat{N}_s^g)\right] \tag{10}$$

Under these notations, we see that $s^* \in \arg\max_{0 \le s \le r} J_{\mathcal{P}}(s; r, \mathcal{D}_{1:n})$ and $s^{\mathrm{us}} \in \arg\max_{0 \le s \le r} \widehat{J}_{\mathcal{P}}(s; r, \widehat{\mathcal{D}}_{1:n})$, where $s^*$ is the optimal decision by $\pi^*$ and $s^{\mathrm{us}}$ is the decision made by our policy $\pi^{\mathrm{our}}$ when given state $(r, \mathcal{D}_{1:n})$.

Let us now define three error quantities that capture population-model misspecification, surrogate approximation error and frontier-level estimation error respectively. For fixed $(r', m) \in \{0, 1, \ldots, r\} \times \{0, 1, \ldots, r\}$, define

$$\Delta_U(r', m) = \left| \mathop{\mathbb{E}}_{\mathcal{D}_{1:m} \sim \mathcal{P}^{\otimes m}} [V_{\mathcal{P}}(r', \mathcal{D}_{1:m})] - U_{\mathcal{P}}(r', m) \right| \tag{11}$$

$$\Delta_{\mathcal{P} \to \widehat{\mathcal{P}}}(r', m) = \left| U_{\mathcal{P}}(r', m) - U_{\widehat{\mathcal{P}}}(r', m) \right| \tag{12}$$

$$\Delta_{\mathcal{D} \to \widehat{\mathcal{D}}}(r, \mathcal{D}_{1:n}) = \sup_{0 \le s \le r} \left| A(s; r, \mathcal{D}_{1:n}) - \widehat{J}_{\mathcal{P}}(s; r, \widehat{\mathcal{D}}_{1:n}) \right| \tag{13}$$

$$A(s; r, \mathcal{D}_{1:n}) = \mathop{\mathbb{E}}_{X_{1:n} \sim \mathcal{D}_{1:n}} \left[ N_s^g + \gamma \cdot U_{\widehat{\mathcal{P}}}(r - s, N_s^g) \right] \tag{14}$$

The following two lemmas Lemma B.2 and Lemma B.3 respectively upper bounds $V_{\mathcal{P}}^{\pi^*}(r, \mathcal{D}_{1:n}) - V_{\mathcal{P}}^{\pi^{\mathrm{our}}}(r, \mathcal{D}_{1:n})$ by $J_{\mathcal{P}}(s^*; r, \mathcal{D}_{1:n}) - J_{\mathcal{P}}(s^{\mathrm{us}}; r, \mathcal{D}_{1:n})$, and then in terms of the the $\Delta$ terms defined above.

**Lemma B.2.** *Fix a state $(r, \mathcal{D}_{1:n})$, population $\mathcal{P}$, and discount factor $\gamma \in (0, 1)$. Let $\pi^*$ be optimal for $V_{\mathcal{P}}$, and let $\pi^{\mathrm{our}}$ be a policy whose first-step decision at $(r, \mathcal{D}_{1:n})$ is a round budget $s^{\mathrm{us}}$, after which it uses the greedy allocation at that budget. Then,*

$$V_{\mathcal{P}}^{\pi^*}(r, \mathcal{D}_{1:n}) - V_{\mathcal{P}}^{\pi^{\mathrm{our}}}(r, \mathcal{D}_{1:n}) \le J_{\mathcal{P}}(s^*; r, \mathcal{D}_{1:n}) - J_{\mathcal{P}}(s^{\mathrm{us}}; r, \mathcal{D}_{1:n})$$

**Lemma B.3.** *Fix a state $(r, \mathcal{D}_{1:n})$, population $\mathcal{P}$, and discount factor $\gamma \in (0, 1)$. Let $\pi^*$ be optimal for $V_{\mathcal{P}}$, and let $\pi^{\mathrm{our}}$ be a policy whose first-step decision at $(r, \mathcal{D}_{1:n})$ is a round budget $s^{\mathrm{us}}$, after which it uses the greedy allocation at that budget. Then,*

$$J_{\mathcal{P}}(s^*; r, \mathcal{D}_{1:n}) - J_{\mathcal{P}}(s^{\mathrm{us}}; r, \mathcal{D}_{1:n}) \le 2\Delta_{\mathcal{D} \to \widehat{\mathcal{D}}}(r, \mathcal{D}_{1:n})$$

$$+ \gamma \cdot \mathop{\mathbb{E}}_{X_{1:n} \sim \mathcal{D}_{1:n}} \left[ \Delta_U(r - s^*, N_{s^*}^g) + \Delta_{\mathcal{P} \to \widehat{\mathcal{P}}}(r - s^*, N_{s^*}^g) + \Delta_U(r - s^{\mathrm{us}}, N_{s^{\mathrm{us}}}^g) + \Delta_{\mathcal{P} \to \widehat{\mathcal{P}}}(r - s^{\mathrm{us}}, N_{s^{\mathrm{us}}}^g) \right]$$

Finally, Theorem 7.2 follows by upper bound each of these $\Delta$ terms with the three lemmas Lemma B.4, Lemma B.5, and Lemma B.6 below.

**Lemma B.4.** *For all $r', m \in \{0, 1, \ldots, r\}$, we have $\Delta_{\mathcal{P} \to \widehat{\mathcal{P}}}(r', m) = |U_{\mathcal{P}}(r', m) - U_{\widehat{\mathcal{P}}}(r', m)| \le \frac{r}{1 - \gamma} \cdot \|\mathcal{P} - \widehat{\mathcal{P}}\|_{\mathrm{TV}}$.*

**Lemma B.5.** *$\Delta_{\mathcal{D} \to \widehat{\mathcal{D}}}(r, \mathcal{D}_{1:n}) = \sup_{0 \le s \le r} \left| A(s; r, \mathcal{D}_{1:n}) - \widehat{J}_{\mathcal{P}}(s; r, \widehat{\mathcal{D}}_{1:n}) \right| \le (1 + \gamma) r \cdot \sum_{i=1}^n \|\mathcal{D}_i - \widehat{\mathcal{D}}_i\|_{\mathrm{TV}}$.*

**Lemma B.6.** *For all $r', m \in \{0, 1, \ldots, r\}$, we have*

$$\Delta_U(r', m) = \left| \mathop{\mathbb{E}}_{\mathcal{D}_{1:m} \sim \mathcal{P}^{\otimes m}} [V_{\mathcal{P}}(r', \mathcal{D}_{1:m})] - U_{\mathcal{P}}(r', m) \right| \le \frac{r^2}{1 - \gamma} \cdot \mathop{\mathbb{E}}_{\mathcal{D} \sim \mathcal{P}} \|\mathcal{D} - \bar{\mathcal{D}}\|_{\mathrm{TV}}$$

*where $\bar{\mathcal{D}} = \mathbb{E}_{\mathcal{D} \sim \mathcal{P}}[\mathcal{D}]$ is the mean distribution of $\mathcal{P}$.*

We are now ready to prove Theorem 7.2.

*Proof.* Combining Lemma B.2, Lemma B.3, Lemma B.5, Lemma B.4, and Lemma B.6, we see that

$$V_{\mathcal{P}}^{\pi^*}(r, \mathcal{D}_{1:n}) - V_{\mathcal{P}}^{\pi^{\mathrm{our}}}(r, \mathcal{D}_{1:n})$$

$$\le J_{\mathcal{P}}(s^*; r, \mathcal{D}_{1:n}) - J_{\mathcal{P}}(s^{\mathrm{us}}; r, \mathcal{D}_{1:n}) \hspace{2cm} \text{(By Lemma B.2)}$$

$$\le 2\Delta_{\mathcal{D} \to \widehat{\mathcal{D}}}(r, \mathcal{D}_{1:n})$$

$$+ \gamma \cdot \mathop{\mathbb{E}}_{X_{1:n} \sim \mathcal{D}_{1:n}} \left[ \Delta_U(r - s^*, N_{s^*}^g) + \Delta_{\mathcal{P} \to \widehat{\mathcal{P}}}(r - s^*, N_{s^*}^g) + \Delta_U(r - s^{\mathrm{us}}, N_{s^{\mathrm{us}}}^g) + \Delta_{\mathcal{P} \to \widehat{\mathcal{P}}}(r - s^{\mathrm{us}}, N_{s^{\mathrm{us}}}^g) \right]$$

$$\hspace{10cm} \text{(By Lemma B.3)}$$

$$\le 2(1 + \gamma) r \cdot \sum_{i=1}^n \|\mathcal{D}_i - \widehat{\mathcal{D}}_i\|_{\mathrm{TV}} + \frac{2\gamma r}{1 - \gamma} \cdot \|\mathcal{P} - \widehat{\mathcal{P}}\|_{\mathrm{TV}} + \frac{2\gamma r^2}{1 - \gamma} \cdot \mathop{\mathbb{E}}_{\mathcal{D} \sim \mathcal{P}} \|\mathcal{D} - \bar{\mathcal{D}}\|_{\mathrm{TV}}$$

$$\hspace{6cm} \text{(By Lemma B.5, Lemma B.4, and Lemma B.6)}$$

This completes the proof. $\qquad\square$

## C. Deferred Proofs

**Proposition C.1** (Marginal decomposition). *For any allocation* $\mathbf{k} \in \mathbb{N}^n$, *we have*

$$\mathbb{E}\left[\sum_{i=1}^{n} \min\{k_i, X_i\}\right] = \sum_{i=1}^{n} \sum_{\ell=1}^{k_i} p_i(\ell)$$

*Proof.* For any $x, y \in \mathbb{N}$, we have

$$\min\{x, y\} = \sum_{\ell=1}^{x} \mathbb{I}[y \geq \ell].$$

where $\mathbb{I}[y \geq \ell]$ is the indicator function of whether $y \geq \ell$. Applying this identity and linearity of expectation, we get

$$\mathbb{E}\left[\sum_{i=1}^{n} \min\{k_i, X_i\}\right] = \mathbb{E}\left[\sum_{i=1}^{n} \sum_{\ell=1}^{k_i} \mathbb{I}[X_i \geq \ell]\right] = \sum_{i=1}^{n} \sum_{\ell=1}^{k_i} \mathbb{E}\left[\mathbb{I}[X_i \geq \ell]\right] = \sum_{i=1}^{n} \sum_{\ell=1}^{k_i} \Pr_{X_i \sim \mathcal{D}_i}(X_i \geq \ell) = \sum_{i=1}^{n} \sum_{\ell=1}^{k_i} p_{\mathcal{D}_i}(\ell)$$

$\square$

**Theorem 4.2** (Optimality of greedy allocation). *For any realized frontier* $\mathcal{D}_{1:n}$ *and any round budget* $s$, *the greedy allocation* $\mathbf{k}^{\text{greedy}}$ *maximizes the single-round objective of* $v_s(\mathcal{D}_{1:n}) = \mathbb{E}_{X_{1:n} \sim \mathcal{D}_{1:n}}[N_s^g]$.

*Proof.* By Proposition 4.1, any allocation $\mathbf{k}$ with $\sum_i k_i \leq s$ achieves expected reward $\sum_{i=1}^{n} \sum_{\ell=1}^{k_i} p_{\mathcal{D}_i}(\ell)$. Equivalently, we select $s$ marginal "slots" $(i, \ell)$ subject to the prefix feasibility constraint: if $(i, \ell)$ is chosen then $(i, 1), \ldots, (i, \ell-1)$ are chosen. Within each individual $i$, the marginal sequence $p_{\mathcal{D}_i}(\ell)$ is non-increasing in $\ell$. Thus, at each step, extending any prefix-feasible partial selection by choosing the currently largest available marginal cannot reduce optimality, and repeating this for $s$ steps yields a maximum-sum prefix-feasible selection. This is exactly the greedy allocation policy. $\square$

**Proposition C.2** (Even allocation is optimal under $\mathcal{P}$). *Fix* $n, s \in \mathbb{N}$, *and define* $a = \lfloor s/n \rfloor$ *and* $c = s - a \cdot n$. *Any optimal solution to* Eq. (4) *satisfies* $k_i \in \{a, a+1\}$ *for all* $i \in [n]$, *with exactly* $c$ *indices satisfying* $k_i = a+1$.

*Proof.* Note $\bar{p}(\ell)$ is non-increasing in $\ell$, hence $g(k+1) - g(k) = \bar{p}(k+1)$ is non-increasing in $k$, and thus $g$ is discrete concave. Fix an optimal allocation $\mathbf{k} = (k_1, \ldots, k_n)$ maximizes $\sum_{i=1}^{n} g(k_i)$ subject to $\sum_{i=1}^{n} k_i = b$.

Suppose, that some pair satisfies $k_i \geq k_j + 2$. Define another valid allocation $\mathbf{k}'$ by shifting one unit from $i$ to $j$:

$$k_z' = \begin{cases} k_z - 1 & \text{if } z = i \\ k_z + 1 & \text{if } z = j \\ k_z & \text{otherwise} \end{cases}$$

Then,

$$\sum_{i=1}^{n} g(k_i') - \sum_{i=1}^{n} g(k_i) = (g(k_j + 1) - g(k_j)) - (g(k_i) - g(k_i - 1)) = \bar{p}(k_j + 1) - \bar{p}(k_i) \geq 0$$

So, $\mathbf{k}'$ is also an optimal allocation that is closer to being an even allocation. Since we can increase the evenness of the allocation without decreasing the objective, iterating this process yields an optimal allocation with all entries differing by at most 1, i.e., in $k_i \in \{q, q+1\}$ for all $i \in [n]$, with exactly $\delta = s - nq$ entries equal to $q + 1$. $\square$

**Theorem 6.2** (Computational complexity of the surrogate). *For a total budget of* $b$, *the population-level value function* $U_{\mathcal{P}}(r, n)$ *can be computed exactly for all* $0 \leq n \leq r \leq b$ *using* $\mathcal{O}(b^2)$ *space and* $\mathcal{O}(b^5 \log b)$ *time.*

*Proof.* We analyze the dynamic program (DP) defined by the surrogate Bellman recursion of Eq. (5) over states $(r, n)$ with $0 \leq n \leq r \leq b$. As there are $\sum_{r=0}^{b}(r+1) \in \mathcal{O}(b^2)$ such states, the DP table requires $\mathcal{O}(b^2)$ space.

Fix a state $(r, n)$ and an action (round budget) $s \in \{0, 1, \ldots, r\}$. Under even allocation with $a = \lfloor \frac{s}{n} \rfloor$ and $c = s - a \cdot n$, the PGF of the next-frontier size $N_s^e$ is

$$T_s(z) = \mathbb{E}[z^{N_s^e}] = \bar{G}_a(z)^{n-c} \cdot \bar{G}_{a+1}(z)^c$$

Since $0 \leq N_s^e \leq s$, it suffices to compute the coefficients of $T_s(z)$ up to degree $s$ by merging all mass above degree $s$ into the coefficient of $z^s$. Using exponentiation-by-squaring with truncation to degree $s$, each exponentiation requires $\mathcal{O}(\log b)$ polynomial multiplications, and each dense multiplication/truncation costs $\mathcal{O}(s^2)$. Hence, computing all coefficients of $T_s$ costs $\mathcal{O}(s^2 \log b)$ time.

Now, given the coefficients $T_s[m] = \Pr(N_s^e = m)$ for $m \in \{0, \dots, s\}$, the expected continuation value is

$$\mathbb{E}\left[U_{\mathcal{P}}(r - s, N_s^e)\right] = \sum_{m=0}^{s} T_s[m] \cdot U_{\mathcal{P}}(r - s, m),$$

which can be computed in $\mathcal{O}(s)$ time. Putting these pieces together, evaluating the Bellman objective for a fixed action $s$ costs $\mathcal{O}(s^2 \log b)$ time, dominated by the polynomial operations.

**Total time complexity.** At state $(r, n)$, we maximize over $s = 0, 1, \dots, r$, so the total cost per state is

$$\sum_{s=0}^{r} \mathcal{O}(s^2 \log b) \subseteq \mathcal{O}(r^3 \log b)$$

Summing over all states yields

$$\sum_{r=0}^{b} \sum_{n=0}^{r} \mathcal{O}(r^3 \log b) \subseteq \sum_{r=0}^{b} \mathcal{O}(r^4 \log b) \subseteq \mathcal{O}(b^5 \log b)$$

as claimed. $\qquad\square$

---

**Algorithm 1** Population-level DP for computing $U_{\mathcal{P}}(r, n)$

---

**Input:** Total budget $b \in \mathbb{N}$, discount factor $\gamma \in (0, 1)$, population model $\mathcal{P}$
**Output:** Table $U[r][n]$, for all $0 \leq n \leq r \leq b$
 1: Compute population-averaged truncated PGFs $\{\bar{G}_k\}_{k=0}^{b}$
 2: Compute population survival probabilities $\{\bar{p}(\ell)\}_{\ell=1}^{b}$ and prefix sums $\left\{ \sum_{\ell=1}^{k} \bar{p}(\ell) \right\}_{k=0}^{b}$
 3: Initialize $U[0][n] = 0$ for all $n \in \{0, \dots, b\}$ and $U[r][0] = 0$ for all $r \in \{0, \dots, b\}$
 4: **for** $r = 1$ **to** $b$ **do**
 5:    **for** $n = 1$ **to** $r$ **do**
 6:       Initialize $U[r][n] = 0$
 7:       **for** $s = 0$ **to** $r$ **do**
 8:          Define $a = \lfloor s/n \rfloor$ and $c \leftarrow s - an$
 9:          Compute $F = n \cdot \sum_{\ell=1}^{a} \bar{p}(\ell) + c \cdot \bar{p}(a+1)$
 10:         Compute $T(z) = \bar{G}_a(z)^{n-c} \cdot \bar{G}_{a+1}(z)^c$, truncated/merged to degree $s$
 11:         Update $U[r][n] = \max\{U[r][n], F + \gamma \cdot \sum_{m=0}^{s} T[m] \cdot U[r-s][m]\}$
 12:       **end for**
 13:    **end for**
 14: **end for**
 15: **return** $U$

---

**Proposition C.3** (Tight single-round error bound). *Fix a round budget $s \in \mathbb{N}$. For any frontier distributions $\mathcal{D}_{1:n}$ and noisy estimates $\widehat{\mathcal{D}}_{1:n}$,*

$$v_s(\mathcal{D}_{1:n}) - v_s(\widehat{\mathcal{D}}_{1:n}) \leq \sum_{i=1}^{n} \sum_{\ell=1}^{s} \left| p_{\mathcal{D}_i}(\ell) - p_{\widehat{\mathcal{D}}_i}(\ell) \right|$$

*Moreover, this bound is tight in the worst case.*

*Proof.* Since we are working in a fixed round $t$, let us suppress the subscript $t$. For each $i \in [n]$ and $\ell \in [s]$, let us define the marginal index set $\mathcal{M}$ and rewrite survival probabilities in terms of pairs of subscripts

$$\mathcal{M} := \{(i, \ell) : i \in [n], \ell \in [s]\}$$
$$p_{(i,\ell)} := p_i(\ell) = \Pr_{X \sim \mathcal{D}_i} (X \geq \ell)$$
$$\widehat{p}_{(i,\ell)} := \widehat{p}_i(\ell) = \Pr_{\widehat{X} \sim \widehat{\mathcal{D}}_i} (\widehat{X} \geq \ell)$$

Any allocation $\mathbf{k} = (k_1, \ldots, k_n)$ with $\sum_{i=1}^{n} k_i = s$ corresponds to the prefix-feasible set

$$\mathcal{S}(\mathbf{k}) := \{(i, \ell) \in \mathcal{M} : 1 \leq \ell \leq k_i\}$$

and by Proposition 4.1, its expected reward equals $\sum_{m \in \mathcal{S}(\mathbf{k})} p_m$.

Let $\mathbf{k}^\star(b)$ be the greedy-optimal allocation computed from the true marginals $\{p_m\}_{m \in \mathcal{M}}$, and let $\widehat{\mathbf{k}}^\star(b)$ be the greedy allocation computed from the noisy marginals $\{\widehat{p}_m\}_{m \in \mathcal{M}}$. Define $\mathcal{S} := \mathcal{S}(\mathbf{k}^\star(b))$ and $\widehat{\mathcal{S}} := \mathcal{S}(\widehat{\mathbf{k}}^\star(b))$ accordingly. Then,

$$\mathbb{E}\left[\sum_{i=1}^{n} \min\{k_i^\star(b), X_i\}\right] = \sum_{m \in \mathcal{S}} p_m \qquad \text{and} \qquad \mathbb{E}\left[\sum_{i=1}^{n} \min\{\widehat{k}_i^\star(b), X_i\}\right] = \sum_{m \in \widehat{\mathcal{S}}} p_m$$

where both expectations are taken over the same underlying realizations $X_i \sim \mathcal{D}_i$.

$$\sum_{m \in \mathcal{S}} p_m - \sum_{m \in \widehat{\mathcal{S}}} p_m = \left(\sum_{m \in \mathcal{S}} (p_m - \widehat{p}_m)\right) + \left(\sum_{m \in \mathcal{S}} \widehat{p}_m - \sum_{m \in \widehat{\mathcal{S}}} \widehat{p}_m\right) + \left(\sum_{m \in \widehat{\mathcal{S}}} (\widehat{p}_m - p_m)\right)$$
$$\text{(Add and subtract the terms } \textstyle\sum_{m \in \mathcal{S}} \widehat{p}_m \text{ and } \sum_{m \in \widehat{\mathcal{S}}} \widehat{p}_m)$$
$$\leq \sum_{m \in \mathcal{S}} (p_m - \widehat{p}_m) + \sum_{m \in \widehat{\mathcal{S}}} (\widehat{p}_m - p_m)$$
$$\text{(Middle term is } \leq 0 \text{ since } \widehat{\mathcal{S}} \text{ maximizes } \textstyle\sum_{m \in \mathcal{S}'} \widehat{p}_m \text{ among prefix-feasible } \mathcal{S}' \text{ of size } s)$$
$$= \sum_{m \in \mathcal{S} \setminus \widehat{\mathcal{S}}} (p_m - \widehat{p}_m) + \sum_{m \in \widehat{\mathcal{S}} \setminus \mathcal{S}} (\widehat{p}_m - p_m) \qquad \text{(Since the intersection cancels each other)}$$
$$\leq \sum_{m \in \mathcal{S} \setminus \widehat{\mathcal{S}}} |p_m - \widehat{p}_m| + \sum_{m \in \widehat{\mathcal{S}} \setminus \mathcal{S}} |\widehat{p}_m - p_m|$$
$$\leq \sum_{m \in \mathcal{M}} |p_m - \widehat{p}_m| \qquad \text{(Since } \mathcal{S} \setminus \widehat{\mathcal{S}} \text{ and } \widehat{\mathcal{S}} \setminus \widehat{\mathcal{S}} \text{ are disjoint subsets of } \mathcal{M})$$
$$= \sum_{i=1}^{n} \sum_{\ell=1}^{s} |p_i(\ell) - \widehat{p}_i(\ell)| \qquad \text{(Definition of } \mathcal{M}; \text{ see above)}$$

**Tightness of the error bound.** Suppose we have a round budget of $s$. Consider disjoint sets $\mathcal{X}$ and $\mathcal{Y}$ over index pairs such that $|\mathcal{X}| = |\mathcal{Y}| = s$. Define $\alpha = \beta/2$. For all $u \in \mathcal{X}$, define $p_u = x$ and $\widehat{p}_u = x - \alpha$. For all $v \in \mathcal{Y}$, define $p_v = x - \beta$ and $\widehat{p}_v = x - \beta + \alpha$. For all other $w \notin \mathcal{X} \cup \mathcal{Y}$, define $p_w = \widehat{p}_w = 0$.

Under $\mathcal{D}$, set $\mathcal{X}$ gets chosen and all vouchers are allocated to first person. Under $\widehat{\mathcal{D}}$, set $\mathcal{Y}$ gets chosen and all vouchers are allocated to second person. Therefore,

$$\sum_{m \in \mathcal{S}} p_m - \sum_{m \in \widehat{\mathcal{S}}} p_m = \sum_{u \in \mathcal{X}} p_u - \sum_{v \in \mathcal{Y}} p_v = s \cdot x - s \cdot (x - \beta) = s \cdot \beta$$

Meanwhile, we see that

$$\sum_{u \in \mathcal{M}} |p_u - \widehat{p}_u| = \sum_{u \in \mathcal{X}} |x - (x - \alpha)| + \sum_{v \in \mathcal{Y}} |(x - \beta) - (x - \beta + \alpha)|$$

(Since $p$ and $\widehat{p}$ only differ on indices within $\mathcal{X} \cup \mathcal{Y}$)

$$= s \cdot \alpha + s \cdot \alpha \qquad \text{(Since } |\mathcal{X}| = |\mathcal{Y}| = s)$$

$$= s \cdot \beta \qquad \text{(Since } \alpha = \beta/2)$$

In other words, we exactly have $\sum_{u \in \mathcal{M}} |p_u - \widehat{p}_u| = \sum_{m \in \mathcal{S}} p_m - \sum_{m \in \widehat{\mathcal{S}}} p_m$. □

## C.1. Proofs for Appendix B

**Lemma C.4.** *Fix a state $(r, \mathcal{D}_{1:n})$, population $\mathcal{P}$, and discount factor $\gamma \in (0, 1)$. Let $\pi^*$ be optimal for $V_\mathcal{P}$, and let $\pi^{\mathrm{our}}$ be a policy whose first-step decision at $(r, \mathcal{D}_{1:n})$ is a round budget $s^{\mathrm{us}}$, after which it uses the greedy allocation at that budget. Then,*

$$V_\mathcal{P}^{\pi^*}(r, \mathcal{D}_{1:n}) - V_\mathcal{P}^{\pi^{\mathrm{our}}}(r, \mathcal{D}_{1:n}) \le J_\mathcal{P}(s^*; r, \mathcal{D}_{1:n}) - J_\mathcal{P}(s^{\mathrm{us}}; r, \mathcal{D}_{1:n})$$

*Proof.* Recall the definition of $J_\mathcal{P}$ from Eq. (9). By optimality of $\pi^*$ and the Bellman equation Eq. (3), we have

$$V_\mathcal{P}^{\pi^*}(r, \mathcal{D}_{1:n}) = V_\mathcal{P}(r, \mathcal{D}_{1:n}) = \max_{0 \le s \le r} J_\mathcal{P}(s; r, \mathcal{D}_{1:n}) = J_\mathcal{P}(s^*; r, \mathcal{D}_{1:n})$$

It remains to show that $V_\mathcal{P}^{\pi^{\mathrm{our}}}(r, \mathcal{D}_{1:n}) \le J_\mathcal{P}(s^{\mathrm{us}}; r, \mathcal{D}_{1:n})$.

Observe that

$$V_\mathcal{P}^{\pi^{\mathrm{our}}}(r, \mathcal{D}_{1:n}) = \underset{X_{1:n} \sim \mathcal{D}_{1:n}}{\mathbb{E}} \left[ N^{\mathrm{us}} + \gamma \cdot \underset{\mathcal{D}'_{1:N^{\mathrm{us}}} | N^{\mathrm{us}}}{\mathbb{E}} \left[ V_\mathcal{P}^{\pi^{\mathrm{our}}}(r - s^{\mathrm{us}}, \mathcal{D}'_{1:N^{\mathrm{us}}}) \right] \right] \qquad \text{(By policy recursion)}$$

$$= \underset{X_{1:n} \sim \mathcal{D}_{1:n}}{\mathbb{E}} \left[ N^g_{s^{\mathrm{us}}} + \gamma \cdot \underset{\mathcal{D}'_{1:N^g_{s^{\mathrm{us}}}} | N^g_{s^{\mathrm{us}}}}{\mathbb{E}} \left[ V_\mathcal{P}(r - s^{\mathrm{us}}, \mathcal{D}'_{1:N^{\mathrm{us}}}) \right] \right]$$

(Since $\pi^{\mathrm{our}}$ uses greedy allocation for budget $s^{\mathrm{us}}$)

$$\le \underset{X_{1:n} \sim \mathcal{D}_{1:n}}{\mathbb{E}} \left[ N^g_{s^{\mathrm{us}}} + \gamma \cdot \underset{\mathcal{D}'_{1:N^{\mathrm{us}}} | N^{\mathrm{us}}}{\mathbb{E}} \left[ V_\mathcal{P}(r - s^{\mathrm{us}}, \mathcal{D}'_{1:N^{\mathrm{us}}}) \right] \right] \qquad (\ddagger)$$

$$= J_\mathcal{P}(s^{\mathrm{us}}; r, \mathcal{D}_{1:n})$$

where $(\ddagger)$ is due to pointwise dominance $V_\mathcal{P}^{\pi^{\mathrm{our}}}(r', \mathcal{D}'_{1:m}) \le V_\mathcal{P}(r', \mathcal{D}'_{1:m})$, for all $(r', \mathcal{D}'_{1:m})$, since $V_\mathcal{P}$ is optimal. □

**Lemma C.5.** *Fix a state $(r, \mathcal{D}_{1:n})$, population $\mathcal{P}$, and discount factor $\gamma \in (0, 1)$. Let $\pi^*$ be optimal for $V_\mathcal{P}$, and let $\pi^{\mathrm{our}}$ be a policy whose first-step decision at $(r, \mathcal{D}_{1:n})$ is a round budget $s^{\mathrm{us}}$, after which it uses the greedy allocation at that budget. Then,*

$$J_\mathcal{P}(s^*; r, \mathcal{D}_{1:n}) - J_\mathcal{P}(s^{\mathrm{us}}; r, \mathcal{D}_{1:n}) \le 2\Delta_{\mathcal{P} \to \widehat{\mathcal{P}}}(r, \mathcal{D}_{1:n})$$

$$+ \gamma \cdot \underset{X_{1:n} \sim \mathcal{D}_{1:n}}{\mathbb{E}} \left[ \Delta_U(r - s^*, N^g_{s^*}) + \Delta_{\mathcal{P} \to \widehat{\mathcal{P}}}(r - s^*, N^g_{s^*}) + \Delta_U(r - s^{\mathrm{us}}, N^g_{s^{\mathrm{us}}}) + \Delta_{\mathcal{P} \to \widehat{\mathcal{P}}}(r - s^{\mathrm{us}}, N^g_{s^{\mathrm{us}}}) \right]$$

*Proof.* For any $s \in \{0, 1, \ldots, r\}$ and any $m \in \{0, 1, \ldots, r - s\}$, we have

$$\left| \underset{\mathcal{D}_{1:m} \sim \mathcal{P}^{\otimes m}}{\mathbb{E}} [V_\mathcal{P}(r - s, \mathcal{D}_{1:m})] - U_{\widehat{\mathcal{P}}}(r - s, m) \right|$$

$$\le \left| \underset{\mathcal{D}_{1:m} \sim \mathcal{P}^{\otimes m}}{\mathbb{E}} [V_\mathcal{P}(r - s, \mathcal{D}_{1:m})] - U_\mathcal{P}(r - s, m) \right| + \left| U_\mathcal{P}(r - s, m) - U_{\widehat{\mathcal{P}}}(r - s, m) \right| \qquad \text{(By triangle inequality)}$$

$$= \Delta_U(r - s, m) + \Delta_{\mathcal{P} \to \widehat{\mathcal{P}}}(r - s, m) \qquad \text{(By Eq. (11) and Eq. (12))}$$

Therefore, for any $s \in \{0, 1, \ldots, r\}$, observe that

$$|J_\mathcal{P}(s; r, \mathcal{D}_{1:n}) - A(s; r, \mathcal{D}_{1:n})| = \left| \gamma \cdot \mathop{\mathbb{E}}_{X_{1:n} \sim \mathcal{D}_{1:n}} \left[ \mathop{\mathbb{E}}_{\mathcal{D}'_{1:N_s^g}|N_s^g} \left[ V_\mathcal{P}(r - s, \mathcal{D}'_{1:N_s^g}) \right] - U_{\widehat{\mathcal{P}}}(r - s, N_s^g) \right] \right| \qquad \text{(By Eq. (9))}$$

$$\leq \gamma \cdot \mathop{\mathbb{E}}_{X_{1:n} \sim \mathcal{D}_{1:n}} \left[ \left| \mathop{\mathbb{E}}_{\mathcal{D}'_{1:N_s^g}|N_s^g} \left[ V_\mathcal{P}(r - s, \mathcal{D}'_{1:N_s^g}) \right] - U_{\widehat{\mathcal{P}}}(r - s, N_s^g) \right| \right]$$

$$\text{(By Jensen's inequality)}$$

$$\leq \gamma \cdot \mathop{\mathbb{E}}_{X_{1:n} \sim \mathcal{D}_{1:n}} \left[ \Delta_U(r - s, m) + \Delta_{\mathcal{P} \to \widehat{\mathcal{P}}}(r - s, m) \right]$$

where the last inequaity is due to conditioning on $N_s^g = m$ and applying the bound above, which holds for any realized $m$. That is, for any $s \in \{0, 1, \ldots, r\}$, we have

$$|J_\mathcal{P}(s; r, \mathcal{D}_{1:n}) - A(s; r, \mathcal{D}_{1:n})| \leq \gamma \cdot \mathop{\mathbb{E}}_{X_{1:n} \sim \mathcal{D}_{1:n}} \left[ \Delta_U(r - s, m) + \Delta_{\mathcal{P} \to \widehat{\mathcal{P}}}(r - s, m) \right] \qquad (15)$$

Meanwhile,

$$A(s^*; r, \widehat{\mathcal{D}}_{1:n}) \leq \widehat{J}_\mathcal{P}(s^*; r, \widehat{\mathcal{D}}_{1:n}) + \Delta_{\mathcal{D} \to \widehat{\mathcal{D}}} \qquad \text{(By Eq. (13))}$$

$$\leq \widehat{J}_\mathcal{P}(s^{\mathrm{us}}; r, \widehat{\mathcal{D}}_{1:n}) + \Delta_{\mathcal{D} \to \widehat{\mathcal{D}}} \qquad \text{(Since } s^{\mathrm{us}} \text{ maximizes } \widehat{J}_\mathcal{P}(s^*; r, \widehat{\mathcal{D}}_{1:n}))$$

$$\leq A(s^{\mathrm{us}}) + 2\Delta_{\mathcal{D} \to \widehat{\mathcal{D}}} \qquad \text{(By Eq. (13))}$$

In other words, we have

$$A(s^*; r, \widehat{\mathcal{D}}_{1:n}) - A(s^{\mathrm{us}}) \leq 2\Delta_{\mathcal{D} \to \widehat{\mathcal{D}}} \qquad (16)$$

Putting together, we see that

$$J_\mathcal{P}(s^*) - J_\mathcal{P}(s^{\mathrm{us}}) = J_\mathcal{P}(s^*) - A(s^*) + A(s^*) - A(s^{\mathrm{us}}) + A(s^{\mathrm{us}}) - J_\mathcal{P}(s^{\mathrm{us}}))$$

$$\text{(Adding and subtracting the terms } A(s^*) \text{ and } A(s^{\mathrm{us}}))$$

$$\leq |J_\mathcal{P}(s^*) - A(s^*)| + A(s^*) - A(s^{\mathrm{us}}) + |J_\mathcal{P}(s^{\mathrm{us}}) - A(s^{\mathrm{us}})| \qquad \text{(Applying absolute values)}$$

$$\leq |J_\mathcal{P}(s^*) - A(s^*)| + 2\Delta_{\mathcal{D} \to \widehat{\mathcal{D}}} + |J_\mathcal{P}(s^{\mathrm{us}}) - A(s^{\mathrm{us}})| \qquad \text{(From Eq. (16))}$$

$$\leq \gamma \cdot \mathop{\mathbb{E}}_{X_{1:n} \sim \mathcal{D}_{1:n}} \left[ \Delta_U(r - s^*, m) + \Delta_{\mathcal{P} \to \widehat{\mathcal{P}}}(r - s^*, m) \right] + 2\Delta_{\mathcal{D} \to \widehat{\mathcal{D}}} + |J_\mathcal{P}(s^{\mathrm{us}}) - A(s^{\mathrm{us}})|$$

$$\text{(From Eq. (15) with } s = s^*)$$

$$\leq \gamma \cdot \mathop{\mathbb{E}}_{X_{1:n} \sim \mathcal{D}_{1:n}} \left[ \Delta_U(r - s^*, m) + \Delta_{\mathcal{P} \to \widehat{\mathcal{P}}}(r - s^*, m) \right] + 2\Delta_{\mathcal{D} \to \widehat{\mathcal{D}}}$$

$$+ \gamma \cdot \mathop{\mathbb{E}}_{X_{1:n} \sim \mathcal{D}_{1:n}} \left[ \Delta_U(r - s^{\mathrm{us}}, m) + \Delta_{\mathcal{P} \to \widehat{\mathcal{P}}}(r - s^{\mathrm{us}}, m) \right] \qquad \text{(From Eq. (15) with } s = s^{\mathrm{us}})$$

$$\square$$

**Lemma C.6.** *For all $r', m \in \{0, 1, \ldots, r\}$, we have $\Delta_{\mathcal{P} \to \widehat{\mathcal{P}}}(r', m) = |U_\mathcal{P}(r', m) - U_{\widehat{\mathcal{P}}}(r', m)| \leq \frac{r}{1-\gamma} \cdot \|\mathcal{P} - \widehat{\mathcal{P}}\|_{\mathrm{TV}}$.*

*Proof.* We consider a general bound $\|U_\mathcal{P}(r', m) - U_{\widehat{\mathcal{P}}}(r', m)\|_\infty$ for any $(r', m) \in \{0, 1, \ldots, r\} \times \{0, 1, \ldots, r\}$. Define the Bellman operator $T_\mathcal{P}$ as

$$T_\mathcal{P} f(x, y) = \max_s \left\{ \mathbb{E}_{\mathcal{D}_{1:y} \sim \mathcal{P}, X_{1:y} \sim \mathcal{D}_{1:y}} [N_s^e + \gamma \cdot f(x - s, N_s^e)] \right\}$$

where $N_s^e = N(\mathbf{k}^{\mathrm{even}}, X_{1:y})$ using $\mathcal{D}_{1:y}$ and $X_{1:y}$ comes from $\mathcal{P}$. Similarly, define $T_{\widehat{\mathcal{P}}}$ as

$$T_{\widehat{\mathcal{P}}} f(x, y) = \max_s \left\{ \mathbb{E}_{\widehat{\mathcal{D}}_{1:y} \sim \widehat{\mathcal{P}}, X_{1:y} \sim \widehat{\mathcal{D}}_{1:y}} [\widehat{N}_s^e + \gamma \cdot f(x - s, \widehat{N}_s^e)] \right\}$$

where $\widehat{N}_s^e = N(\mathbf{k}^{\mathrm{even}}, X_{1:y})$ using $\widehat{\mathcal{D}}_{1:y}$ and $X_{1:y}$ comes from $\widehat{\mathcal{P}}$.

Thus, $T_{\mathcal{P}} u_{\mathcal{P}}(r', m) = u_{\mathcal{P}}(r', m)$ and $T_{\widehat{\mathcal{P}}} u_{\widehat{\mathcal{P}}}(r', m) = u_{\widehat{\mathcal{P}}}(r', m)$. Then we have

$$\|u_{\mathcal{P}}(r', m) - u_{\widehat{\mathcal{P}}}(r', m)\|_\infty = \|T_{\mathcal{P}} u_{\mathcal{P}}(r', m) - T_{\widehat{\mathcal{P}}} u_{\widehat{\mathcal{P}}}(r', m)\|_\infty \qquad \text{(By fixed points of Bellman operator)}$$
$$\leq \|T_{\mathcal{P}} u_{\mathcal{P}}(r', m) - T_{\mathcal{P}} u_{\widehat{\mathcal{P}}}(r', m)\|_\infty + \|T_{\mathcal{P}} u_{\widehat{\mathcal{P}}}(r', m) - T_{\widehat{\mathcal{P}}} u_{\widehat{\mathcal{P}}}(r', m)\|_\infty.$$
$$\text{(By triangle inequality)}$$

We separately bound the two terms on the right hand side.

**Bounding the first term:** From the contraction property of the Bellman operator, we have:

$$\|T_{\mathcal{P}} u_{\mathcal{P}}(r', m) - T_{\mathcal{P}} u_{\widehat{\mathcal{P}}}(r', m)\|_\infty \leq \gamma \cdot \|u_{\mathcal{P}}(r', m) - u_{\widehat{\mathcal{P}}}(r', m)\|_\infty$$

**Bounding the second term:** We bound the error due to model misspecification by explicitly tracing the sampling process $\mathcal{P} \to \mathcal{D}_{1:m} \to X_{1:m} \to N_s^e = N(\mathbf{k}^{\text{even}}, X_{1:m})$. Let $u(n, s) = n + \gamma \cdot u_{\widehat{\mathcal{P}}}(r' - s, n)$ denote the value term. We use $g_s(n)$ to denote the distribution of $N_s^e$ induced by $\mathcal{P}$, and $\widehat{g}_s(n)$ to denote the distribution of $\widehat{N}_s^e$ induced by $\widehat{\mathcal{P}}$.

$$\|T_{\mathcal{P}} u_{\widehat{\mathcal{P}}}(r', m) - T_{\widehat{\mathcal{P}}} u_{\widehat{\mathcal{P}}}(r', m)\|_\infty$$

$$= \left\|\max_s \mathop{\mathbb{E}}_{\substack{\mathcal{D}_{1:m} \sim \mathcal{P} \\ X_{1:m} \sim \mathcal{D}_{1:m}}} [u(N(\mathbf{k}_s^{\text{even}}, X_{1:m}), s)] - \max_s \mathop{\mathbb{E}}_{\substack{\widehat{\mathcal{D}}_{1:m} \sim \widehat{\mathcal{P}} \\ X_{1:m} \sim \widehat{\mathcal{D}}_{1:m}}} [u(N(\mathbf{k}_s^{\text{even}}, X_{1:m}), s)] \right\|_\infty$$

$$\leq \max_{r', m, s} \left| \mathop{\mathbb{E}}_{\substack{\mathcal{D}_{1:m} \sim \mathcal{P} \\ X_{1:m} \sim \mathcal{D}_{1:m}}} [u(N_s^e, s)] - \mathop{\mathbb{E}}_{\substack{\widehat{\mathcal{D}}_{1:m} \sim \widehat{\mathcal{P}} \\ X_{1:m} \sim \widehat{\mathcal{D}}_{1:m}}} [u(\widehat{N}_s^e, s)] \right| \qquad (|\max f - \max g| \leq \max |f - g|)$$

$$= \max_{r', m, s} \left| \sum_{n=1}^s g_s(n) \cdot u(n, s) - \sum_{n=1}^s \widehat{g}_s(n) \cdot u(n, s) \right| \qquad \text{(Rewrite expectation using } g_s \text{ and } \widehat{g}_s)$$

$$\leq \max_{r', m, s} \left( \max_{0 \leq n \leq s} |u(n, s)| \right) \cdot \|g_s - \widehat{g}_s\|_{\text{TV}} \qquad \text{(Definition of TV distance)}$$

$$\leq r \cdot \|g_s - \widehat{g}_s\|_{\text{TV}} \qquad \text{(since } |u(n, s)| \leq n + \gamma \cdot (r' - s) \leq s + \gamma(r - s) \leq r)$$

$$\leq r \cdot \|\mathcal{P} - \widehat{\mathcal{P}}\|_{\text{TV}} \qquad \text{(Data Processing Inequality)}$$

For the last inequality, since $g_s$ and $\widehat{g}_s$ are obtained by applying the same sampling and deterministic calculation process to $\mathcal{P}$ and $\widehat{\mathcal{P}}$ respectively, the distance between the outputs is bounded by the distance between the inputs.

Combining the bounds above, we obtain:

$$\|u_{\mathcal{P}}(r', m) - u_{\widehat{\mathcal{P}}}(r', m)\|_\infty \leq \gamma \|u_{\mathcal{P}}(r', m) - u_{\widehat{\mathcal{P}}}(r', m)\|_\infty + r \cdot \|\mathcal{P} - \widehat{\mathcal{P}}\|_{\text{TV}}.$$

The claim follows by rearranging the terms to solve for $\|u_{\mathcal{P}} - u_{\widehat{\mathcal{P}}}\|_\infty$. $\qquad \square$

**Lemma C.7.** $\Delta_{\mathcal{D} \to \widehat{\mathcal{D}}}(r, \mathcal{D}_{1:n}) = \sup_{0 \leq s \leq r} \left| A(s; r, \mathcal{D}_{1:n}) - \widehat{J}_{\mathcal{P}}(s; r, \widehat{\mathcal{D}}_{1:n}) \right| \leq (1 + \gamma) r \cdot \sum_{i=1}^n \|\mathcal{D}_i - \widehat{\mathcal{D}}_i\|_{\text{TV}}.$

*Proof.* Recall the definition of $A$ and $\widehat{J}_{\mathcal{P}}$:

$$\widehat{J}_{\widehat{\mathcal{P}}}(s; r, \widehat{\mathcal{D}}_{1:n}) := \mathop{\mathbb{E}}_{\widehat{X}_{1:n} \sim \widehat{\mathcal{D}}_{1:n}} \left[ \widehat{N}_s^g + \gamma \cdot U_{\widehat{\mathcal{P}}}(r - s, \widehat{N}_s^g) \right]$$

$$A(s; r, \mathcal{D}_{1:n}) := \mathop{\mathbb{E}}_{X_{1:n} \sim \mathcal{D}_{1:n}} \left[ N_s^g + \gamma \cdot U_{\widehat{\mathcal{P}}}(r - s, N_s^g) \right]$$

where $N_s^g = N(\mathbf{k}^{\text{greedy}}; X_{1:n})$ and $\widehat{N}_s^g := N(\widehat{\mathbf{k}}^{\text{greedy}}; \widehat{X}_{1:n})$, with allocation $\mathbf{k}^{\text{greedy}}$ is computed from $\mathcal{D}_{1:n}$ and allocation

$\widehat{\mathbf{k}}^{\text{greedy}}$ is computed from $\widehat{\mathcal{D}}_{1:n}$. By triangle inequality and Proposition 7.1, we get

$$
\begin{aligned}
&|\widehat{J}_{\widehat{\mathcal{P}}}(s; r, \widehat{\mathcal{D}}_{1:n}) - A(s; r, \mathcal{D}_{1:n})| \\
&\leq \left| \mathop{\mathbb{E}}_{X_{1:n} \sim \mathcal{D}_{1:n}} [N_s^g] - \mathop{\mathbb{E}}_{\widehat{X}_{1:n} \sim \widehat{\mathcal{D}}_{1:n}} [\widehat{N}_s^g] \right| + \gamma \cdot \left| \mathop{\mathbb{E}}_{\widehat{X}_{1:n} \sim \widehat{\mathcal{D}}_{1:n}} [U_{\mathcal{P}}(r-s, \widehat{N}_s^g)] - \mathop{\mathbb{E}}_{X_{1:n} \sim \mathcal{D}_{1:n}} [U_{\mathcal{P}}(r-s, N_s^g)] \right| \\
&\hspace{10cm} \text{(By triangle inequality)} \\
&\leq \sum_{i=1}^n \sum_{l=1}^s |p_i(\ell) - \widehat{p}_i(\ell)| + \gamma \cdot \left| \mathop{\mathbb{E}}_{\widehat{X}_{1:n} \sim \widehat{\mathcal{D}}_{1:n}} [U_{\mathcal{P}}(r-s, \widehat{N}_s^g)] - \mathop{\mathbb{E}}_{X_{1:n} \sim \mathcal{D}_{1:n}} [U_{\mathcal{P}}(r-s, N_s^g)] \right| \quad \text{(By Proposition 7.1)}
\end{aligned}
$$

Thus, it remains to bound $\left| \mathbb{E}_{\widehat{X}_{1:n} \sim \widehat{\mathcal{D}}_{1:n}} [U_{\mathcal{P}}(r-s, \widehat{N}_s^g)] - \mathbb{E}_{X_{1:n} \sim \mathcal{D}_{1:n}} [U_{\mathcal{P}}(r-s, N_s^g)] \right|$.

Let us define two intermediate variables $M_s^g = N(\mathbf{k}^{\text{greedy}}; \widehat{X}_{1:n})$ and $\widehat{M}_s^g = N(\widehat{\mathbf{k}}^{\text{greedy}}; X_{1:n})$. From Theorem 4.2, we know that $\mathbf{k}^{\text{greedy}}$ is the optimal assignment for $\mathcal{D}_{1:n}$, so $\mathbb{E}_{X_{1:n} \sim \mathcal{D}_{1:n}} [U_{\mathcal{P}}(r-s, \widehat{M}_s^g)] \leq \mathbb{E}_{X_{1:n} \sim \mathcal{D}_{1:n}} [U_{\mathcal{P}}(r-s, N_s^g)]$. Similarly, we have $\mathbb{E}_{\widehat{X}_{1:n} \sim \widehat{\mathcal{D}}_{1:n}} [U_{\mathcal{P}}(r-s, M_s^g)] \leq \mathbb{E}_{\widehat{X}_{1:n} \sim \widehat{\mathcal{D}}_{1:n}} [U_{\mathcal{P}}(r-s, \widehat{N}_s^g)]$. Thus,

$$
\begin{aligned}
&\left| \mathop{\mathbb{E}}_{\widehat{X}_{1:n} \sim \widehat{\mathcal{D}}_{1:n}} [U_{\mathcal{P}}(r-s, \widehat{N}_s^g)] - \mathop{\mathbb{E}}_{X_{1:n} \sim \mathcal{D}_{1:n}} [U_{\mathcal{P}}(r-s, N_s^g)] \right| \\
&\leq \max \left\{ \left| \mathop{\mathbb{E}}_{\widehat{X}_{1:n} \sim \widehat{\mathcal{D}}_{1:n}} [U_{\mathcal{P}}(r-s, \widehat{N}_s^g)] - \mathop{\mathbb{E}}_{X_{1:n} \sim \mathcal{D}_{1:n}} [U_{\mathcal{P}}(r-s, \widehat{M}_s^g)] \right|, \left| \mathop{\mathbb{E}}_{X_{1:n} \sim \mathcal{D}_{1:n}} [U_{\mathcal{P}}(r-s, N_s^g)] - \mathop{\mathbb{E}}_{\widehat{X}_{1:n} \sim \widehat{\mathcal{D}}_{1:n}} [U_{\mathcal{P}}(r-s, M_s^g)] \right| \right\}
\end{aligned}
$$

since $a \geq c$ and $b \geq d$ implies that $|a - b| \leq \max\{|a - d|, |b - c|\}$. It remains to individually bound each of the terms in the maximization, where each of them is the absolute difference between two terms with the *same* allocation $\mathbf{k}$.

Let's start by bounding $\left| \mathbb{E}_{X_{1:n} \sim \mathcal{D}_{1:n}} [U_{\mathcal{P}}(r-s, N_s^g)] - \mathbb{E}_{\widehat{X}_{1:n} \sim \widehat{\mathcal{D}}_{1:n}} [U_{\mathcal{P}}(r-s, M_s^g)] \right|$. Let $g(n)$ denote the distribution of $N_s^g$ and $g_i(n)$ to denote the distribution of $\min\{\mathbf{k}_i^{\text{greedy}}, X_i\}$, where $N_s^g = N(\mathbf{k}^{\text{greedy}}; X_{1:n}) = \sum_{i=1}^n \min\{\mathbf{k}_i^{\text{greedy}}, X_i\}$. Then,

$$
\begin{aligned}
&\left| \mathop{\mathbb{E}}_{X_{1:n} \sim \mathcal{D}_{1:n}} [U_{\mathcal{P}}(r-s, N_s^g)] - \mathop{\mathbb{E}}_{\widehat{X}_{1:n} \sim \widehat{\mathcal{D}}_{1:n}} [U_{\mathcal{P}}(r-s, M_s^g)] \right| \\
&= \left| \sum_{n=1}^s g(n) \cdot U_{\mathcal{P}}(r-s, n) - \sum_{n=1}^s h(n) \cdot U_{\mathcal{P}}(r-s, n) \right| \qquad \text{(Rewrite expectation using } g(n) \text{ and } h(n)) \\
&\leq \left| \max_{1 \leq n \leq s} U_{\mathcal{P}}(r-s, n) \right| \cdot \|g - h\|_{\text{TV}} \qquad\qquad\qquad\qquad \text{(Definition of TV distance)} \\
&\leq (r-s) \cdot \|g - h\|_{\text{TV}} \qquad\qquad\qquad\qquad\qquad\qquad\qquad \text{(Since } U_{\mathcal{P}}(a, \cdot) \leq a) \\
&\leq (r-s) \cdot \sum_{i=1}^n \|g_i - h_i\|_{\text{TV}} \qquad\qquad\qquad\qquad\qquad \text{(Subadditivity of TV distance)} \\
&\leq (r-s) \cdot \sum_{i=1}^n \|\mathcal{D}_i - \widehat{\mathcal{D}}_i\|_{\text{TV}} \qquad\qquad\qquad\qquad\qquad \text{(Data Processing Inequality)}
\end{aligned}
$$

Now, let $h(n)$ denote the distribution of $M_s^g$ and $h_i(n)$ to denote the distribution of $\min\{\mathbf{k}_i^{\text{greedy}}, \widehat{X}_i\}$, where $M_s^g = N(\mathbf{k}^{\text{greedy}}; \widehat{X}_{1:n}) = \sum_{i=1}^n \min\{\mathbf{k}_i^{\text{greedy}}, \widehat{X}_i\}$. We can repeat the same argument as above, with $g$ and $g_i$ replaced by $h$ and $h_i$ respectively, and conclude that

$$
\left| \mathop{\mathbb{E}}_{X_{1:n} \sim \mathcal{D}_{1:n}} [U_{\mathcal{P}}(r-s, \widehat{M}_s^g)] - \mathop{\mathbb{E}}_{\widehat{X}_{1:n} \sim \widehat{\mathcal{D}}_{1:n}} [U_{\mathcal{P}}(r-s, \widehat{N}_s^g)] \right| \leq (r-s) \cdot \sum_{i=1}^n \|\mathcal{D}_i - \widehat{\mathcal{D}}_i\|_{\text{TV}}.
$$

Combining all terms together, we have

$$\Delta_{\mathcal{D}\to\widehat{\mathcal{D}}}(r, \mathcal{D}_{1:n}) \leq \sup_{0 \leq s \leq r} \left( \sum_{i=1}^{n} \sum_{l=1}^{r} |p_i(\ell) - \widehat{p}_i(\ell)| + (r-s)\gamma \cdot \sum_{i=1}^{n} \|\mathcal{D}_i - \widehat{\mathcal{D}}_i\|_{\mathrm{TV}} \right) \qquad \text{(From above)}$$

$$\leq (1+\gamma)r \cdot \sum_{i=1}^{n} \|\mathcal{D}_i - \widehat{\mathcal{D}}_i\|_{\mathrm{TV}} \qquad \text{(Since } |p_i(\ell) - \widehat{p}_i(\ell)| \leq \|\mathcal{D}_i - \widehat{\mathcal{D}}_i\|_{\mathrm{TV}} \text{ for all } \ell \in [r])$$

which concludes the proof. □

**Lemma C.8.** *For all $r', m \in \{0, 1, \ldots, r\}$, we have*

$$\Delta_U(r', m) = \left| \mathop{\mathbb{E}}_{\mathcal{D}_{1:m} \sim \mathcal{P}^{\otimes m}} [V_{\mathcal{P}}(r', \mathcal{D}_{1:m})] - U_{\mathcal{P}}(r', m) \right| \leq \frac{r^2}{1-\gamma} \cdot \mathop{\mathbb{E}}_{\mathcal{D} \sim \mathcal{P}} \|\mathcal{D} - \bar{\mathcal{D}}\|_{\mathrm{TV}}$$

*where $\bar{\mathcal{D}} = \mathbb{E}_{\mathcal{D} \sim \mathcal{P}}[\mathcal{D}]$ is the mean distribution of $\mathcal{P}$.*

*Proof.* Recall that

$$V_{\mathcal{P}}(r, \mathcal{D}_{1:m}) = \max_{0 \leq s \leq r} \left\{ \mathop{\mathbb{E}}_{X_{1:m} \sim \mathcal{D}_{1:m}} \left[ N_s^g + \gamma \cdot \mathop{\mathbb{E}}_{\mathcal{D}'_{1:N_s^g} \mid N_s^g} \left[ V_{\mathcal{P}}(r - s, \mathcal{D}'_{1:N_s^g}) \right] \right] \right\}$$

and

$$U_{\mathcal{P}}(r, m) = \max_{0 \leq s \leq r} \mathop{\mathbb{E}}_{\mathcal{D}_{1:m} \sim \mathcal{P}^{\otimes m}} \mathop{\mathbb{E}}_{X_{1:m} \sim \mathcal{D}_{1:n}} [N_s^e + \gamma \cdot U_{\mathcal{P}}(r - s, N_s^e)].$$

We now define the expected version of $V_{\mathcal{P}}(r, \mathcal{D}_{1:m})$:

$$W_{\mathcal{P}}(r, m) = \mathop{\mathbb{E}}_{\mathcal{D}_{1:m} \sim \mathcal{P}^{\otimes m}} \left[ \max_{0 \leq s \leq r} \left\{ \mathop{\mathbb{E}}_{X_{1:m} \sim \mathcal{D}_{1:m}} \left[ N_s^g + \gamma \cdot \mathop{\mathbb{E}}_{\mathcal{D}'_{1:N_s^g} \mid N_s^g} \left[ V_{\mathcal{P}}(r - s, \mathcal{D}'_{1:N_s^g}) \right] \right] \right\} \right],$$

and $\Delta_U(r', m)$ can be written as $|W_{\mathcal{P}}(r', m) - U_{\mathcal{P}}(r', m)|$.

Similar to the proof of Lemma B.4, we define two Bellman operator $T^g$ and $T^e$:

$$(T^g f)(x, y) := \mathop{\mathbb{E}}_{\mathcal{D}_{1:y} \sim \mathcal{P}^{\otimes y}} \left[ \max_{0 \leq s \leq x} \mathop{\mathbb{E}}_{X_{1:y} \sim \mathcal{D}_{1:y}} [N_s^g + \gamma f(x - s, N_s^g)] \right],$$

where $N_s^g = N(\mathbf{k}^{\mathrm{greedy}}, X_{1:y})$. Also,

$$(T^e f)(x, y) := \max_{0 \leq s \leq x} \mathop{\mathbb{E}}_{\mathcal{D}_{1:y} \sim \mathcal{P}^{\otimes y}} \left[ \mathop{\mathbb{E}}_{X_{1:y} \sim \mathcal{D}_{1:y}} [N_s^e + \gamma f(x - s, N_s^e)] \right],$$

where $N_s^e = N(\mathbf{k}^{\mathrm{even}}, X_{1:y})$. Thus, $T^g W_{\mathcal{P}}(r', m) = W_{\mathcal{P}}(r', m)$ and $T^e U_{\mathcal{P}}(r', m) = U_{\mathcal{P}}(r', m)$. Then we have

$$\|W_{\mathcal{P}}(r', m) - U_{\mathcal{P}}(r', m)\|_\infty = \|T^g W_{\mathcal{P}}(r', m) - T^e U_{\mathcal{P}}(r', m)\|_\infty \qquad \text{(By fixed points of Bellman operator)}$$

$$\leq \|T^g W_{\mathcal{P}}(r', m) - T^g U_{\mathcal{P}}(r', m)\|_\infty + \|T^g U_{\mathcal{P}}(r', m) - T^e U_{\mathcal{P}}(r', m)\|_\infty.$$
$$\text{(By triangle inequality)}$$

We separately bound the two terms on the right hand side.

**Bounding the first term:** From the contraction property of the Bellman operator, we have:

$$\|T^g W_{\mathcal{P}}(r', m) - T^g U_{\mathcal{P}}(r', m)\|_\infty \leq \gamma \|W_{\mathcal{P}}(r', m) - U_{\mathcal{P}}(r', m)\|_\infty$$

**Bounding the second term:** For $r', m \in \{0, 1, \ldots, r\}$, define the following terms

$$F_s(\mathcal{D}_{1:m}) := \mathop{\mathbb{E}}_{X_{1:m} \sim \mathcal{D}_{1:m}} [N_s^g + \gamma U_{\mathcal{P}}(r' - s, N_s^g)]$$

$$H_s(\mathcal{P}) := \mathop{\mathbb{E}}_{\mathcal{D}_{1:m} \sim \mathcal{P}^{\otimes m}} \mathop{\mathbb{E}}_{X'_{1:m} \sim \mathcal{D}_{1:m}} [M_s^g + \gamma \cdot U_{\mathcal{P}}(r' - s, M_s^g)]$$

where $M_s^g = N(\mathbf{k}^{\text{greedy}}, X'_{1:m})$ uses the same allocation $\mathbf{k}^{\text{greedy}}$ but on the population-distribution $\mathcal{P}$. By definition of $T^g$ and $T^e$, we have $T^g U_{\mathcal{P}}(r', m) = \max_{0 \leq s \leq m} F_s(\mathcal{D}_{1:m})$ and $T^e U_{\mathcal{P}}(r', m) = \max_{0 \leq s \leq m} H_s(\mathcal{P})$.

$$
\begin{aligned}
&\|T^g U_{\mathcal{P}}(r', m) - T^e U_{\mathcal{P}}(r', m)\|_\infty \\
&= \max_{r', m \in \{0, 1, \ldots, r\}} (T^g U_{\mathcal{P}}(r', m) - T^e U_{\mathcal{P}}(r', m)) && \text{(For any } f \text{ and } (x, y), T^g f(x, y) \geq T^e f(x, y)) \\
&= \max_{r', m \in \{0, 1, \ldots, r\}} \left( \mathop{\mathbb{E}}_{\mathcal{D}_{1:m} \sim \mathcal{P}^{\otimes m}} \max_{0 \leq s \leq r} F_s(\mathcal{D}_{1:m}) - \max_s H_s(\mathcal{P}) \right) && \text{(Definition of } T^g \text{ and } T^e) \\
&\leq \max_{r', m \in \{0, 1, \ldots, r\}} \mathop{\mathbb{E}}_{\mathcal{D}_{1:m} \sim \mathcal{P}^{\otimes m}} \left[ \max_{0 \leq s \leq r} (F_s(\mathcal{D}_{1:m}) - H_s(\mathcal{P})) \right] && (\max f - \max g \leq \max(f - g))
\end{aligned}
$$

Thus, it suffices to upper bound $F_s(\mathcal{D}_{1:m}) - H_s(\mathcal{P})$ for any $0 \leq s \leq r$. To do so, let $f(n)$ denote the distribution of $N_s^g$ and $f_i(n)$ denote the distribution of $\min\{\mathbf{k}^{\text{greedy}}, X_i\}$. Similarly, let $h(n)$ denote the distribution of $M_s^g$ and $h_i(n)$ denote the distribution of $\min\{\mathbf{k}^{\text{greedy}}, X'_i\}$. Further, let $u(n, s) = n + \gamma \cdot U_{\mathcal{P}}(r' - s, n)$ denote the value term. Then, we have

$$
\begin{aligned}
F_s(\mathcal{D}_{1:m}) - H_s(\mathcal{P}) &\leq |F_s(\mathcal{D}_{1:m}) - H_s(\mathcal{P})| \\
&= \left| \sum_{n=1}^s f(n) \cdot u(n, s) - h(n) \cdot u(n, s) \right| && \text{(Rewrite expectation using } f \text{ and } h) \\
&\leq \left( \max_{1 \leq n \leq s} |u(n, s)| \right) \cdot \|f - h\|_{\text{TV}} && \text{(Definition of TV distance)} \\
&\leq (s + \gamma(r' - s)) \cdot \sum_{n=1}^s |f(n) - h(n)| && \text{(Since } |u(n, s)| \leq n + \gamma \cdot (r' - s) \leq s + \gamma(r' - s)) \\
&\leq (s + \gamma(r' - s)) \cdot \sum_{i=1}^m \|f_i - h_i\|_{\text{TV}} && \text{(Subadditivity of TV distance))} \\
&\leq (s + \gamma(r' - s)) \cdot \sum_{i=1}^m \|\mathcal{D}_i - \bar{\mathcal{D}}\|_{\text{TV}} && \text{(Data Processing Inequality)}
\end{aligned}
$$

So,

$$
\begin{aligned}
\|T^g U_{\mathcal{P}}(r', m) - T^e U_{\mathcal{P}}(r', m)\|_\infty &\leq \max_{r', m \in \{0, 1, \ldots, r\}} \mathop{\mathbb{E}}_{\mathcal{D}_{1:m} \sim \mathcal{P}^{\otimes m}} \left[ \max_{0 \leq s \leq r} (F_s(\mathcal{D}_{1:m}) - H_s(\mathcal{P})) \right] && \text{(From above)} \\
&\leq \max_{r', m \in \{0, 1, \ldots, r\}} \mathop{\mathbb{E}}_{\mathcal{D}_{1:m} \sim \mathcal{P}^{\otimes m}} \left[ r \cdot \sum_{i=1}^m \|\mathcal{D}_i - \bar{\mathcal{D}}\|_{\text{TV}} \right] && \text{(Since } s + \gamma(r' - s) \leq r) \\
&\leq r^2 \cdot \mathop{\mathbb{E}}_{\mathcal{D} \sim \mathcal{P}} \left[ \|\mathcal{D} - \bar{\mathcal{D}}\|_{\text{TV}} \right] && \text{(Homogeneity of } \mathcal{D}_i \text{ in expectation)}
\end{aligned}
$$

**Combining the bounds on both terms:** From above, we have

$$
\begin{aligned}
\|W_{\mathcal{P}}(r', m) - U_{\mathcal{P}}(r', m)\|_\infty &\leq \|T^g W_{\mathcal{P}}(r', m) - T^g U_{\mathcal{P}}(r', m)\|_\infty + \|T^g U_{\mathcal{P}}(r', m) - T^e U_{\mathcal{P}}(r', m)\|_\infty \\
&\leq \gamma \cdot \|W_{\mathcal{P}}(r', m) - U_{\mathcal{P}}(r', m)\|_\infty + r^2 \cdot \mathop{\mathbb{E}}_{\mathcal{D} \sim \mathcal{P}} \left[ \|\mathcal{D} - \bar{\mathcal{D}}\|_{\text{TV}} \right]
\end{aligned}
$$

Therefore,

$$
\begin{aligned}
\left| \mathop{\mathbb{E}}_{\mathcal{D}_{1:m} \sim \mathcal{P}^{\otimes m}} [V_{\mathcal{P}}(r', \mathcal{D}_{1:m})] - U_{\mathcal{P}}(r', m) \right| &= |W_{\mathcal{P}}(r', m) - U_{\mathcal{P}}(r', m)| \\
&\leq \|W_{\mathcal{P}}(r', m) - U_{\mathcal{P}}(r', m)\|_\infty \\
&\leq \frac{r^2}{1 - \gamma} \cdot \mathop{\mathbb{E}}_{\mathcal{D} \sim \mathcal{P}} \left[ \|\mathcal{D} - \bar{\mathcal{D}}\|_{\text{TV}} \right]
\end{aligned}
$$

as claimed. $\qquad \square$

# D. Experimental Details

This appendix provides additional details on experimental design, datasets, and implementation choices. The main paper reports representative configurations for clarity, while the results here document the full range of settings explored.

**Interpretation of the resource allocation model.**    Throughout our experiments, we interpret the budget as a collection of identical resources such as referral vouchers, self-test kits, or incentives. Allocating $k_i$ units of budget to an individual $i$ corresponds to giving that individual up to $k_i$ opportunities to recruit peers. Each individual can utilize only a random subset of these resources, reflecting heterogeneous willingness, availability, or opportunity to recruit others. This abstraction aligns closely with real-world respondent-driven sampling and incentivized testing campaigns, where only a fraction of distributed resources lead to effective downstream recruitment.

**ICPSR dataset and empirical recruitment networks.**    For real-world experiments, we use a de-identified, public-use dataset released by ICPSR (Morris & Rothenberg, 2011).[3] The dataset was originally collected to study how social and partnership networks influence the transmission of sexually transmitted and blood-borne infections. It contains social contact networks together with demographic covariates (e.g., gender, housing status, employment status) and reported disease status for multiple infections, including Gonorrhea, Chlamydia, Syphilis, HIV, and Hepatitis.

In our experiments, nodes correspond to individuals and edges correspond to potential recruitment pathways. When simulating recruitment, allocating $k_i$ resources to an individual corresponds to selecting up to $k_i$ neighbors not yet recruited. This setting allows us to evaluate policies both under stochastic referral realizations drawn from fitted distributions and under realized network dynamics.

**Constructing empirical referral distributions.**    To construct a population distribution $\mathcal{P}$ over referral distributions from the ICPSR data, we group individuals with similar covariates using a decision-tree-based partitioning. Specifically, we fit a decision tree that predicts observed degree from covariates, and treat each leaf as a population subgroup. Within each subgroup, we estimate an empirical distribution over observed degrees, yielding a collection of referral distributions. Sampling from $\mathcal{P}$ then corresponds to first sampling a subgroup according to its empirical frequency, and then sampling a referral distribution from that subgroup. This procedure induces a population distribution consistent with the modeling assumptions of Section 3 while preserving interpretable heterogeneity.

**Population-level distribution approximation.**    To compute the population-level surrogate value function, we require access to the marginal distribution induced by drawing an individual distribution $\mathcal{D} \sim \mathcal{P}$ and then drawing a referral count $X \sim \mathcal{D}$. We approximate this mixture distribution using Monte Carlo sampling from $\mathcal{P}$. Specifically, for a fixed tail threshold $\tau$, we estimate $\mathbb{E}_{\mathcal{D} \sim \mathcal{P}} \left[ \Pr_{X \sim \mathcal{D}}(X = j) \right]$ for $j = 0, 1, \ldots, \tau - 1$ and aggregate all remaining probability mass into a single tail bucket $\Pr(X \geq \tau) = 1 - \sum_{j=0}^{\tau-1} \Pr(X = j)$. This truncated representation preserves total probability mass and yields an exact probability generating function up to degree $\tau$. In all experiments, we choose $\tau$ to be at least the maximum feasible round budget, ensuring that truncation does not affect optimal allocation decisions or transition probabilities used by the surrogate dynamic program.

**Implementation details.**    All surrogate value functions are computed using the truncated population-level dynamic program described in Section 6. Polynomial operations are implemented with explicit truncation to control computational complexity. Experiments are implemented in Python and executed on a shared computing cluster using Slurm job arrays. Full code and scripts used to generate all results are available on Github.[4]

## D.1. Additional plots

Here, we present the plots for other diseases — Chlamydia, Gonorrhea, Hepatitis, and Syphilis — in the ICPSR dataset.

---

[3]The dataset is publicly available at https://www.icpsr.umich.edu/web/ICPSR/studies/22140 upon agreeing to ICPSR's terms of use. As a de-identified public-use dataset, it does not require IRB approval.

[4]https://github.com/cxjdavin/Adaptive-Multi-Round-Allocation-with-Stochastic-Arrivals

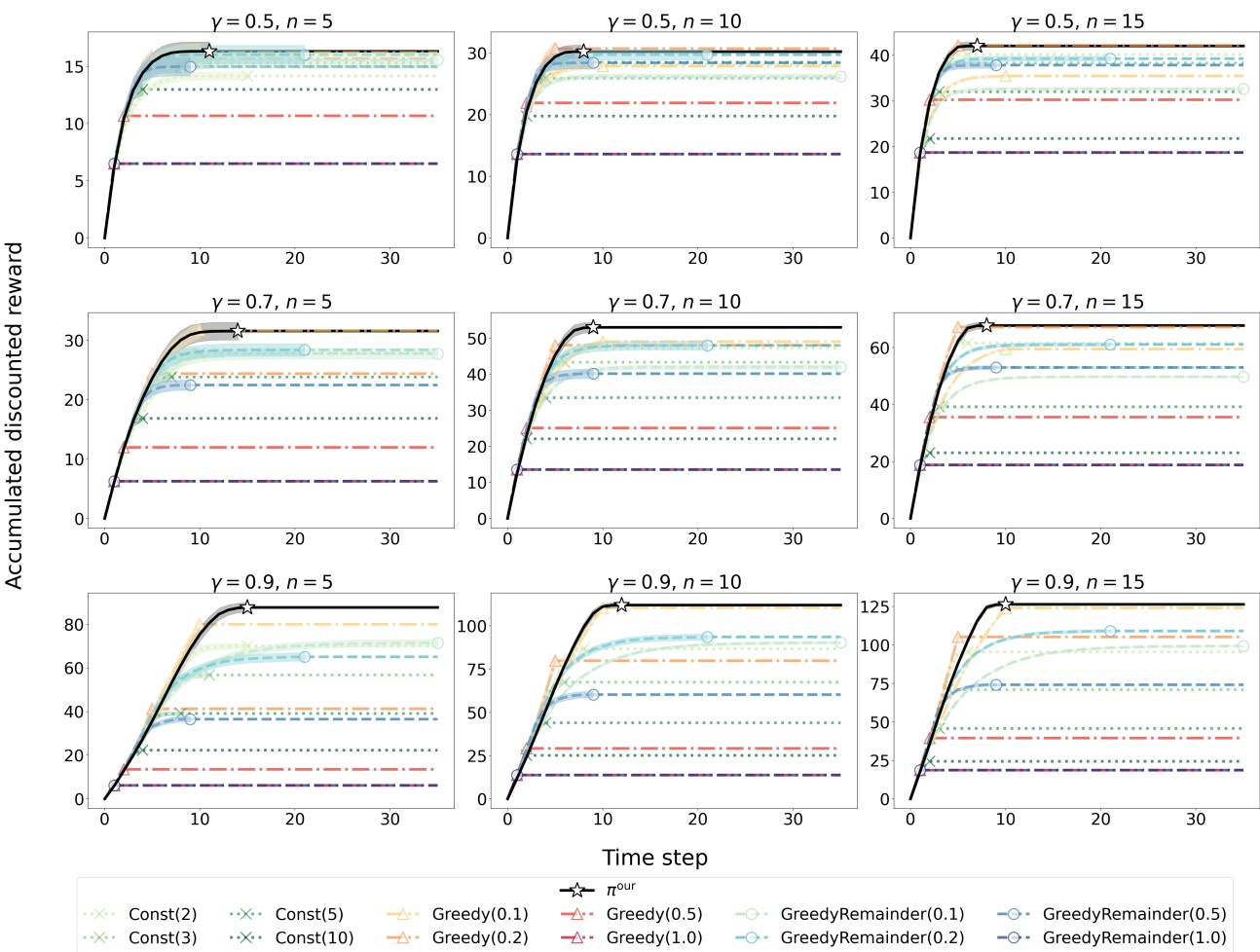

*Figure 5.* Experimental plots for simulated referrals from ICPSR Chlamydia network. We compare our proposed policy $\pi^{\mathrm{our}}$ (solid black line) against the constant- and greedy-allocation baselines described in Section 8 for discount factors $\gamma \in \{0.5, 0.7, 0.9\}$ and initial frontier size $n \in \{5, 10, 15\}$. Each configuration is evaluated over 30 independent runs and we report mean accumulated discounted reward with standard error bands. Markers indicate the termination of the recruitment process, either due to budget exhaustion or an empty frontier.

*Figure 6.* Experimental plots for realized referrals from ICPSR Chlamydia network. We compare our proposed policy $\pi^{\mathrm{our}}$ (solid black line) against the constant- and greedy-allocation baselines described in Section 8 for discount factors $\gamma \in \{0.5, 0.7, 0.9\}$ and initial frontier size $n \in \{5, 10, 15\}$. Each configuration is evaluated over 30 independent runs and we report mean accumulated discounted reward with standard error bands. Markers indicate the termination of the recruitment process, either due to budget exhaustion or an empty frontier.

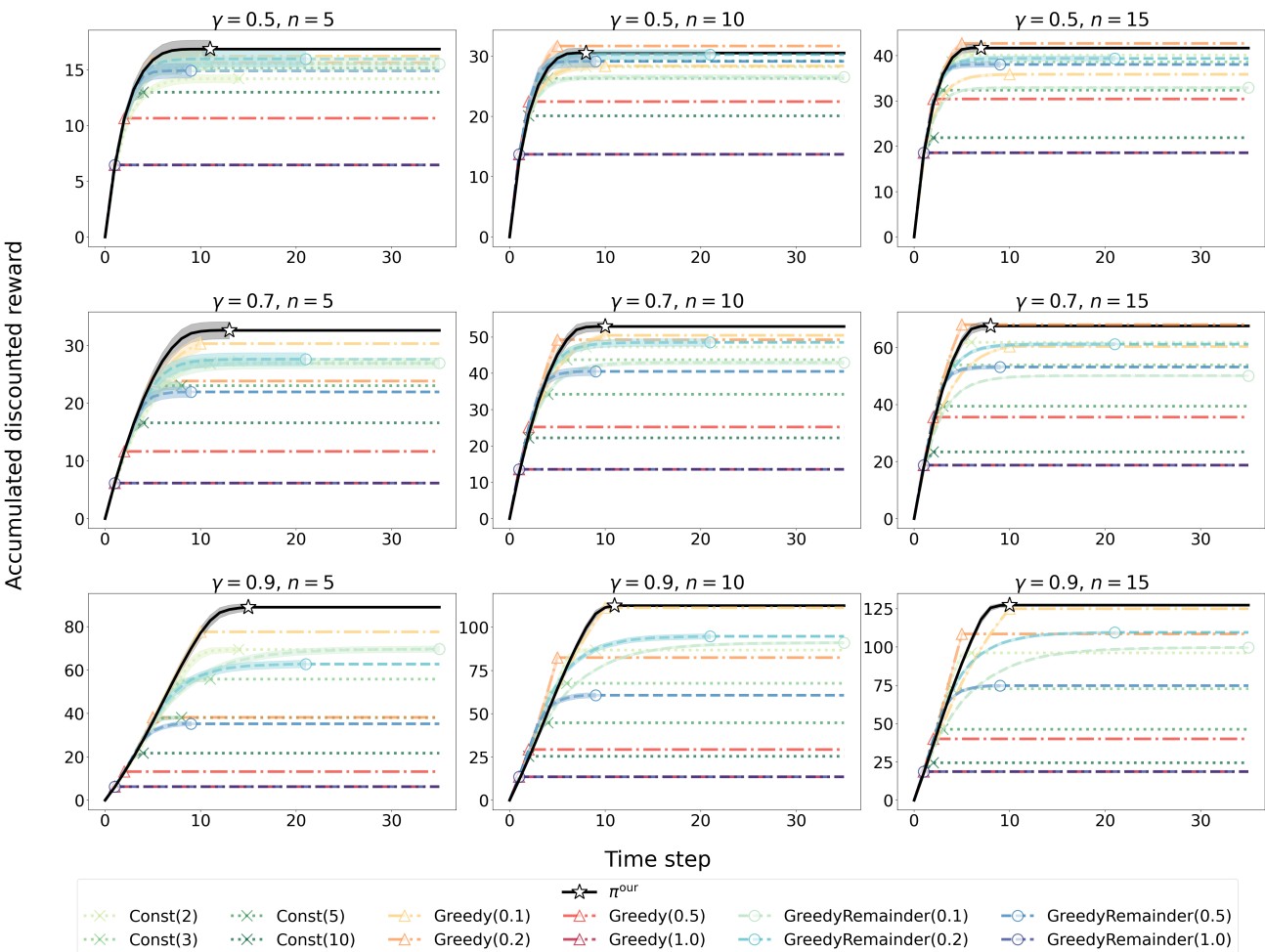

*Figure 7.* Experimental plots for simulated referrals from ICPSR Gonorrhea network. We compare our proposed policy $\pi^{\mathrm{our}}$ (solid black line) against the constant- and greedy-allocation baselines described in Section 8 for discount factors $\gamma \in \{0.5, 0.7, 0.9\}$ and initial frontier size $n \in \{5, 10, 15\}$. Each configuration is evaluated over 30 independent runs and we report mean accumulated discounted reward with standard error bands. Markers indicate the termination of the recruitment process, either due to budget exhaustion or an empty frontier.

*Figure 8.* Experimental plots for realized referrals from ICPSR Gonorrhea network. We compare our proposed policy $\pi^{\text{our}}$ (solid black line) against the constant- and greedy-allocation baselines described in Section 8 for discount factors $\gamma \in \{0.5, 0.7, 0.9\}$ and initial frontier size $n \in \{5, 10, 15\}$. Each configuration is evaluated over 30 independent runs and we report mean accumulated discounted reward with standard error bands. Markers indicate the termination of the recruitment process, either due to budget exhaustion or an empty frontier.

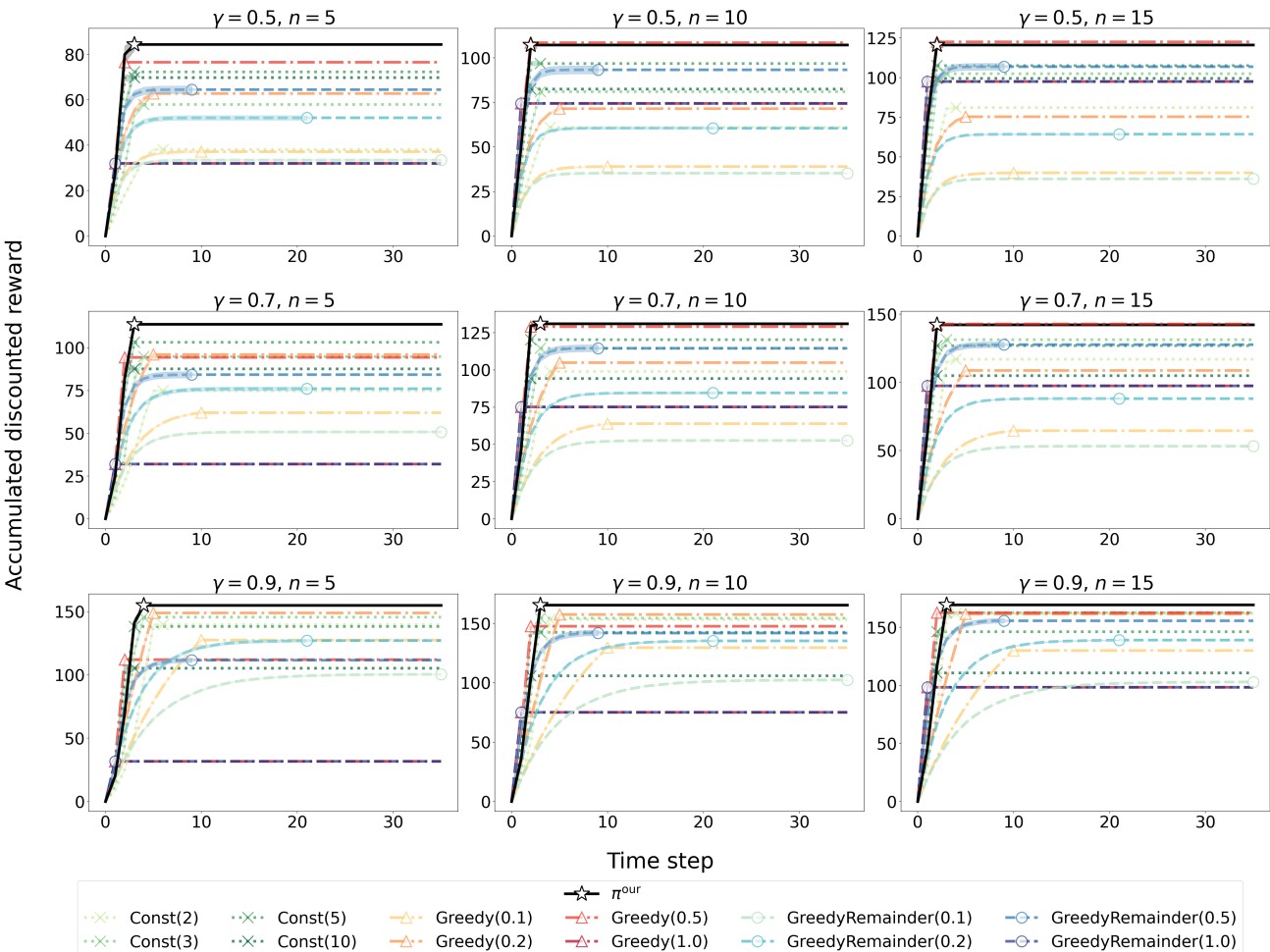

*Figure 9.* Experimental plots for simulated referrals from ICPSR Hepatitis network. We compare our proposed policy $\pi^{\text{our}}$ (solid black line) against the constant- and greedy-allocation baselines described in Section 8 for discount factors $\gamma \in \{0.5, 0.7, 0.9\}$ and initial frontier size $n \in \{5, 10, 15\}$. Each configuration is evaluated over 30 independent runs and we report mean accumulated discounted reward with standard error bands. Markers indicate the termination of the recruitment process, either due to budget exhaustion or an empty frontier.

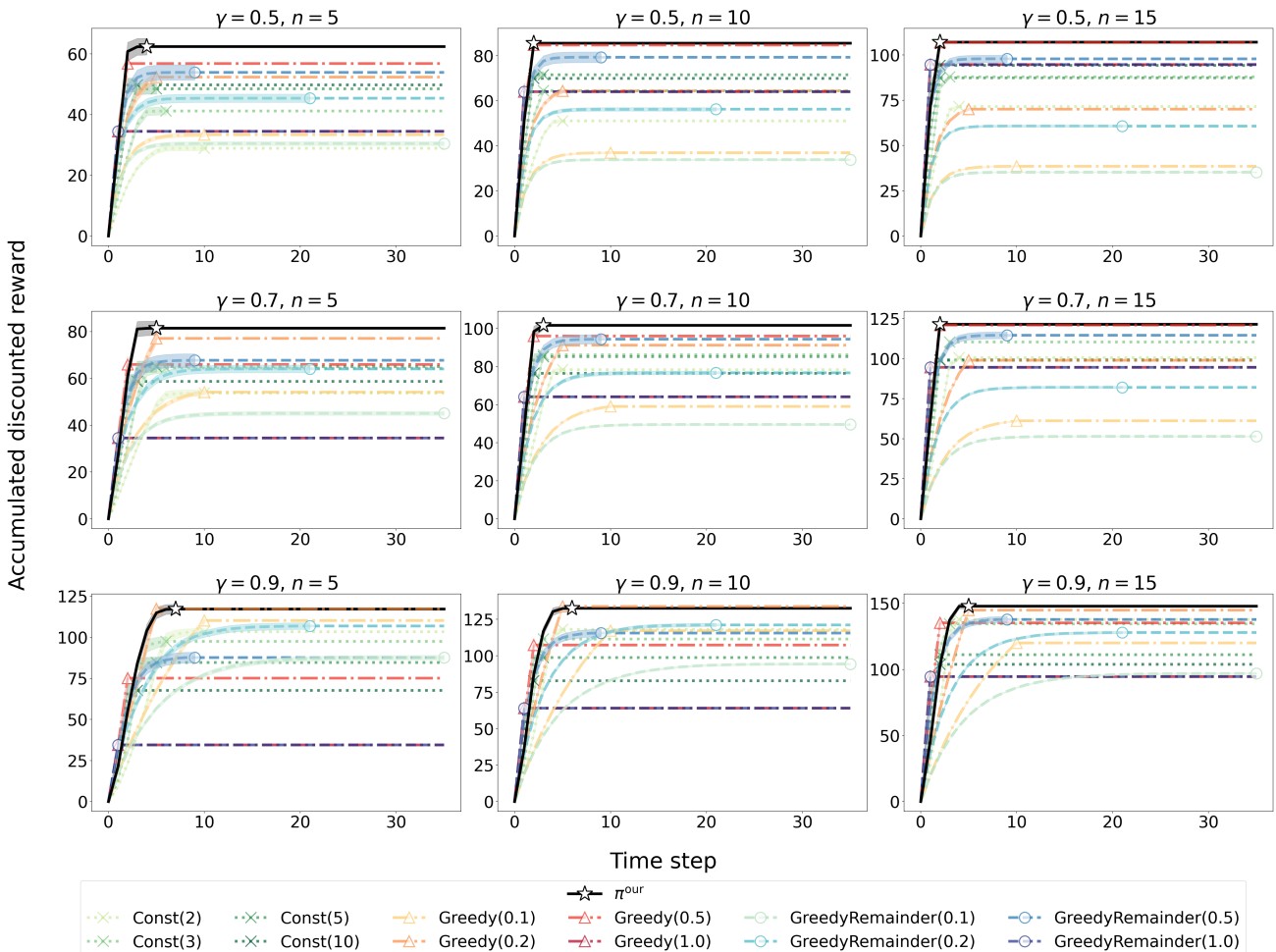

*Figure 10.* Experimental plots for realized referrals from ICPSR Hepatitis network. We compare our proposed policy $\pi^{\mathrm{our}}$ (solid black line) against the constant- and greedy-allocation baselines described in Section 8 for discount factors $\gamma \in \{0.5, 0.7, 0.9\}$ and initial frontier size $n \in \{5, 10, 15\}$. Each configuration is evaluated over 30 independent runs and we report mean accumulated discounted reward with standard error bands. Markers indicate the termination of the recruitment process, either due to budget exhaustion or an empty frontier.

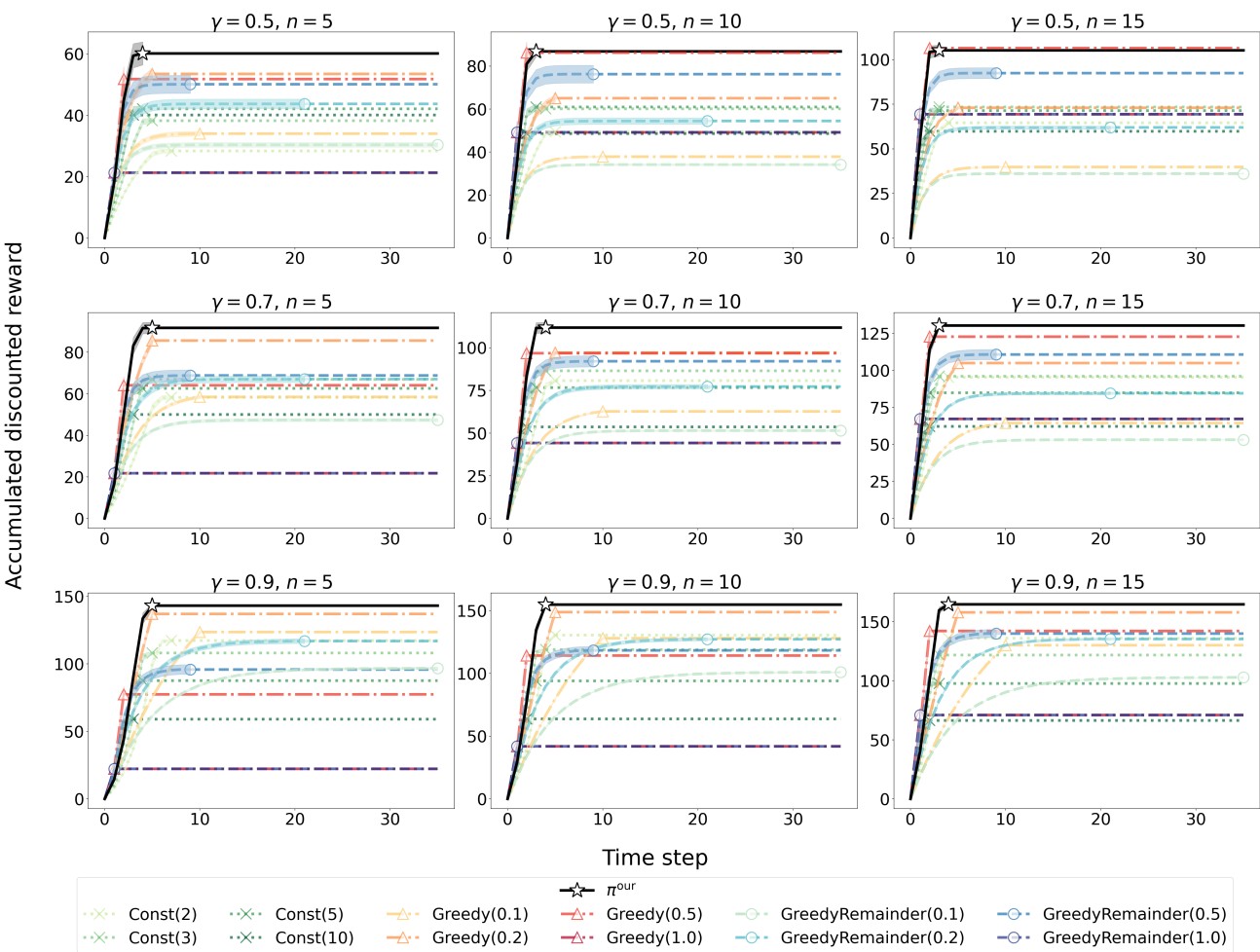

*Figure 11.* Experimental plots for simulated referrals from ICPSR Syphilis network. We compare our proposed policy $\pi^{\text{our}}$ (solid black line) against the constant- and greedy-allocation baselines described in Section 8 for discount factors $\gamma \in \{0.5, 0.7, 0.9\}$ and initial frontier size $n \in \{5, 10, 15\}$. Each configuration is evaluated over 30 independent runs and we report mean accumulated discounted reward with standard error bands. Markers indicate the termination of the recruitment process, either due to budget exhaustion or an empty frontier.

*Figure 12.* Experimental plots for realized referrals from ICPSR Syphilis network. We compare our proposed policy $\pi^{\mathrm{our}}$ (solid black line) against the constant- and greedy-allocation baselines described in Section 8 for discount factors $\gamma \in \{0.5, 0.7, 0.9\}$ and initial frontier size $n \in \{5, 10, 15\}$. Each configuration is evaluated over 30 independent runs and we report mean accumulated discounted reward with standard error bands. Markers indicate the termination of the recruitment process, either due to budget exhaustion or an empty frontier.

