# OpenReview forum: "Adaptive Multi-Round Allocation with Stochastic Arrivals"
_ICML.cc/2026/Conference — ICML 2026 regular_

### Official Review · Reviewer_umWA · 2026-03-03

**Soundness:** 4
**Presentation:** 3
**Significance:** 3
**Originality:** 3
**Overall Recommendation:** 4
**Confidence:** 4

**Summary:**

This paper studies a multi-round, budget-constrained resource allocation problem motivated by adaptive network recruitment. In each round, a decision-maker allocates limited resources to individuals with stochastic referral capacities. Successful referrals generate immediate reward and also create future decision opportunities, so allocation decisions affect both current outcomes and the size of future frontiers. This leads to a stochastic control problem with endogenous and variable-dimensional state evolution.

The authors first show that the single-round allocation problem admits an exact greedy solution based on marginal survival probabilities. This result relies on a clean decomposition of the expected reward into diminishing marginal contributions, which makes the greedy rule optimal within a round. They then formulate the multi-round problem as a dynamic program. Since the exact Bellman equation is intractable due to the distribution-valued state and random frontier growth, they introduce a population-level surrogate value function that depends only on the remaining budget and frontier size. This surrogate allows them to construct an exact dynamic program using truncated probability generating functions, with polynomial computational complexity. The paper also provides robustness guarantees under noisy estimates of individual-level and population-level referral distributions, yielding a multi-round error bound that decomposes into frontier estimation error, population misspecification, and a heterogeneity-induced surrogate approximation term. Experiments on synthetic and real-world recruitment settings demonstrate improvements over constant-allocation baselines and robustness to moderate noise.

**Compliance With Llm Reviewing Policy:**

Affirmed.

**Key Questions For Authors:**

The surrogate-based policy relies on collapsing the frontier state to its size while ignoring composition. Do you have any structural conditions under which this surrogate can be shown to be near-optimal relative to the true multi-round optimal policy? Even a bound under restricted heterogeneity or light-tailed distributions would strengthen the theoretical contribution. If such a result is possible, it would increase my confidence in the broader applicability of the method.

The computational complexity of the surrogate dynamic program is polynomial but relatively high. Could you comment on how the method scales in practice as the total budget grows? Are there natural approximations or pruning strategies that preserve most of the performance while reducing computational cost?

In the experiments, the main comparisons are against constant-allocation policies. Have you considered comparing against stronger adaptive baselines, such as myopic greedy without population-level planning or simple rollout heuristics? Seeing how your approach compares to other adaptive strategies would help clarify the practical gains from the surrogate planning component.

The framework assumes i.i.d. sampling of arrival distributions from a population model. In real recruitment networks, correlations or clustering effects may arise. Could you comment on how sensitive the approach might be to violations of this independence assumption?

**Limitations:**

The authors do discuss limitations in the impact statement and acknowledge modeling assumptions. That said, it would be helpful to more explicitly emphasize a few technical limitations in the main text. In particular, the lack of a worst-case approximation guarantee for the surrogate-based policy relative to the true optimal multi-round solution is worth highlighting. The approach is supported by stability-style bounds, but it remains unclear how large the gap could be in highly heterogeneous settings.

It would also be useful to discuss scalability more explicitly, given the polynomial but high computational complexity, and to comment on how robust the method is to structural deviations from the i.i.d. population model. Overall, the paper is thoughtful and careful, but a slightly more explicit discussion of these practical and theoretical boundaries would strengthen it further.

**Strengths And Weaknesses:**

Overall, the paper is technically solid and clearly organized. The single-round greedy result is clean and not obvious, and the way the authors separate within-round allocation from across-round budget planning makes sense and is easy to follow. The population-level surrogate is a smart way to simplify a very complex state space while still keeping the key dynamics of the problem. The use of probability generating functions to compute transitions exactly is neat and well done. The robustness analysis is also helpful. The error bound breaks down the different sources of performance loss in a way that is easy to interpret. In the experiments, the results line up with the theory and show clear gains, especially in heavy-tailed settings where differences between individuals matter more.

There are also some limitations. Although the paper gives stability guarantees under estimation error, it does not show how close the surrogate-based policy is to the true optimal multi-round policy in the worst case. In particular, there is no constant-factor or clear near-optimality guarantee under general heterogeneity. The method is polynomial-time, but the running time is still fairly high, which could make it hard to scale to larger budgets. The experiments mainly compare against constant-allocation policies. While these are common in practice, they are relatively simple baselines, and it would be helpful to see comparisons with stronger adaptive methods. Finally, the model assumes independence and i.i.d. sampling from a population distribution, and it is not clear how sensitive the approach is if these assumptions do not hold in real network settings.

Overall, this is a careful and well-executed paper on a difficult stochastic control problem. It offers clear structural insights and solid technical work. Stronger optimality guarantees and broader empirical comparisons would make the contribution even stronger.

---

> ### Author Rebuttal · Authors · 2026-03-30
>
> We appreciate your time and effort in reviewing our work. We address your concerns below; please let us know if you require further clarifications. Thank you!
>
> # When is the surrogate a good approximation (Q1)
> Our current analysis already gives a partial structural answer to this question.  In Theorem 7.2, the suboptimality of our policy relative to the optimal policy is decomposed into three terms: frontier-estimation error, population-model misspecification, and an intrinsic surrogate error term. If we consider the correctly specified case where the first two terms vanish, the remaining gap is controlled by the heterogeneity term $E_{\mathcal{D}\sim\mathcal{P}}\Vert\mathcal{D}-\bar{\mathcal{D}}\Vert_{\mathrm{TV}}$, which is exactly the intrinsic error from collapsing composition to frontier size. Thus, under restricted heterogeneity, when $\mathcal{P}$ is centralized near its mean distribution $\bar{\mathcal{D}}$, our surrogate is near-optimal, and in the homogeneous case the bound becomes zero. Our present analysis does not use tail assumptions, so we do not currently prove a sharper light-tailed guarantee. Deriving such results for structured distribution families would be an interesting extension.
>
> # Computational concern (Q2)
> The $O(b^5 \log b)$ complexity in Theorem 6.2 is a **one-time offline cost** for precomputing the surrogate table $U_P(r,n)$ over all states $(r,n)$. Once this table is built, online execution only requires solving a single-round greedy allocation together with evaluating the precomputed surrogate. We will clarify this offline/online distinction explicitly in the revision.
>
> On a 2024 MacBook Pro with M4 chip, $b\in\\{100,150,200\\}$ took about 200, 530, and 1500 seconds to compute the surrogate respectively. For larger budgets, a practical heuristic is to solve the surrogate for a smaller budget then scale up proportionally. Additionally, one can show that $U$ is concave so one can implement a binary search styled algorithm to reduce the complexity to $O(b^4 \log b)$, though we did not do so in our current implementation.
>
> # Experimental concerns (Q3)
> In practice, RDS implementations almost universally impose a fixed number of coupons per participant (typically 2–5) [3,4,5,6], as recommended in WHO operational guidelines [8]. This corresponds directly to constant-allocation policies in our model. Thus, constant allocation is not a weak baseline, but the standard policy used in practice.
>
> That said, we agree with the reviewers that more baselines are useful. Building on Reviewer 127x's suggestion, we tested two additional classes of policies: a non-adaptive policy $\texttt{Greedy}(\alpha)$ and an adaptive policy $\texttt{GreedyRemainder}(\alpha)$
>
> (i) $\texttt{Greedy}(\alpha)$ applies single-round greedy allocation with fixed budget $\alpha b$ per round
>
> (ii) $\texttt{GreedyRemainder}(\alpha)$ allocates a fraction $\alpha$ of the remaining budget each round.
>
> We evaluate both across different values of $\alpha\in [0,1]$, yielding ten additional baselines. Note that $\texttt{Greedy}(1) = \texttt{GreedyRemainder}(1)$ simply exhausts the entire budget in the first round. On real-world recruitment networks, our proposed method consistently outperforms these baselines on real-world recruitment networks (see the plot in https://github.com/icml2026authors/Rebuttal-for-Adaptive-Multi-Round-Allocation-with-Stochastic-Arrivals/blob/main/rebuttal.png), demonstrating the benefit of multi-round planning beyond myopic allocation.
>
> For intuition, **we proved that greedy is optimal for any given per-round budget**. If the optimal cross-round budget schedule were known, greedy under that schedule would be optimal. Thus, these baselines are strong comparators: they isolate the effect of budget scheduling while using optimal within-round decisions. Our gains therefore arise from a better state-dependent allocation schedule across rounds.
>
> # Our modeling choice (Q4)
> Please see our response to Reviewer F1GZ ("Our modeling choice").
>
> # References
>
> [3] McCreesh, Nicky, et al. Respondent Driven Sampling: Determinants of Recruitment and a Method to Improve Point Estimation. PLoS One 8.10 (2013): e78402.
>
> [4] Wylie, John L., and Ann M. Jolly. Understanding recruitment: outcomes associated with alternate methods for seed selection in respondent driven sampling. BMC Medical Research Methodology 13.1 (2013): 93.
>
> [5] Truong, Hong-Ha M., et al. Respondent-driven sampling to recruit adolescents in Kenya. Annals of Epidemiology 78 (2023): 68-73.
>
> [6] Wang, W., Sun, G., Li, C., Qiu, C., Fan, J. and Jin, Y. Exploring the mechanism of trait depression and cognitive impairment on the formation of among individuals with methamphetamine use disorder under varying degrees of social support. Frontiers in Public Health, 13:1435511, 2025.
>
> [8] World Health Organization. Introduction to Respondent-Driven Sampling. World Health Organization, 2013. Available at https://applications.emro.who.int/dsaf/emrpub_2013_en_1539.pdf.

---

> > ### Author Rebuttal · Reviewer_umWA · 2026-03-31
> >
> > Thank you for your detailed response. I appreciate the clarification and will keep my score as is.

---

> > > ### Author Response · Authors · 2026-04-06
> > >
> > > We sincerely appreciate your follow-up and are glad that our response has adequately addressed your concerns! Thank you again for your positive recommendation and thoughtful feedback, and we hope you can recommend our work for acceptance.

---

### Official Review · Reviewer_127x · 2026-03-07

**Soundness:** 4
**Presentation:** 4
**Significance:** 3
**Originality:** 3
**Overall Recommendation:** 4
**Confidence:** 4

**Summary:**

This paper studies sequential resource allocation for network recruitment. In each round, the system observes a "frontier" of eligible individuals, each characterized by features that induce a probability distribution $\mathcal{D}$ over their potential referral capacity. The system must determine both the total budget $s$ for the current round and the specific allocation $k_i$ for each individual to maximize the total (discounted) number of recruits generated till the process ends (either budget exhausts or no individuals in frontier).
The authors show that the single round allocation problem admits an exact greedy solution based on marginal survival probabilities. They then use this for the multi-round allocation, but the exact multi-round Bellman recursion is computationally intractable. To resolve this, they introduce a surrogate value function that depends only on the remaining budget and frontier size. The authors also provide theoretical error bounds for settings where referral distributions must be estimated from data. Their method is evaluated against constant-allocation baselines on synthetic populations and real-world contact networks.

**Compliance With Llm Reviewing Policy:**

Affirmed.

**Key Questions For Authors:**

- The population model assumes that while individual $\mathcal{D}_i$ are observed, and future recruits are characterized only by the prior $\mathcal{P}$. How would the surrogate construction change if recruitment was modeled as a functional form (e.g., $f(\text{covariates}) + \epsilon$)?,

**Limitations:**

Yes

**Strengths And Weaknesses:**

Strengths
- The paper is well-organized and they show an elegant decomposition, reducing the multi-round problem into a sequence of greedy allocations.
- Section 7 provides a stability analysis of the Bellman recursion under noise and the multi-round error decomposition in Theorem 7.2 is valuable to understand how frontier estimation errors propagate.


Weaknesses
- Wouldn't the $b^5$ factor would run into issues for large budgets, I see you use b=200 in your experiments, also is this incurred in each round or for the whole sequence?
- In Section 2 you discuss the connections to Branching MDPs, Stochastic Knapsacks, and Bandits with Knapsacks, but these are not utilized as benchmarks in the experiments. The main comparison is against a constant-allocation policy ($Const(k)$), which does not utilize the distributional information ($\mathcal{D}_{1:n}$) available, and can be weak benchmark?
- Did you try other benchmarks other than constant-allocation? 1) What about only using the one-Step Greedy (myopic) policy
or 2) An adaptive baseline like a fraction policy where $s = \alpha(t) \times \text{remaining budget}$, and you solve the one step greedy with this budget.

---

> ### Author Rebuttal · Authors · 2026-03-30
>
> We appreciate your time and effort in reviewing our work. We address your concerns below; please let us know if you require further clarifications. Thank you!
>
> # Computational concern (W1)
>
> The $O(b^5 \log b)$ complexity in Theorem 6.2 is a **one-time offline cost** for precomputing the surrogate table $U_P(r,n)$ over all states $(r,n)$. Once this table is built, online execution only requires solving a single-round greedy allocation together with evaluating the precomputed surrogate. We will clarify this offline/online distinction explicitly in the revision.
>
> On a 2024 MacBook Pro with M4 chip, $b \in \\{100, 150, 200\\}$ took about 200, 530, and 1500 seconds to compute the surrogate respectively. For larger budgets, a practical heuristic is to solve the surrogate for a smaller budget then scale up proportionally. Additionally, one can show that $U$ is concave so one can implement a binary search styled algorithm to reduce the complexity to $O(b^4 \log b)$, though we did not do so in our current implementation.
>
> # Related work (W2)
>
> The connections to branching MDPs, stochastic knapsack, and bandits with knapsacks are intended to position our problem within a broader class of resource-constrained sequential decision problems. However, these formulations do not yield directly comparable policies for our setting. The key difference is that in our model (see right half of Lines 92-99 in our paper), current allocation decisions **endogenously generate future decision opportunities** by determining the next frontier, whereas in the above models the future state evolution is not shaped in this way by current actions. For this reason, they are conceptually related but not directly applicable as experimental baselines.
>
> # Experimental concerns (W3)
>
> In practice, RDS implementations almost universally impose a fixed number of coupons per participant (typically 2–5) [3,4,5,6], as recommended in WHO operational guidelines [8]. This corresponds directly to constant-allocation policies in our model. Thus, constant allocation is not a weak baseline, but the standard policy used in practice.
>
> That said, we agree with the reviewers that more baselines are useful. Building on your suggestion, we tested two additional classes of policies: a non-adaptive policy $\texttt{Greedy}(\alpha)$ and an adaptive policy $\texttt{GreedyRemainder}(\alpha)$
>
> (i) $\texttt{Greedy}(\alpha)$ applies single-round greedy allocation with fixed budget $\alpha b$ per round
>
> (ii) $\texttt{GreedyRemainder}(\alpha)$ allocates a fraction $\alpha$ of the remaining budget each round.
>
> We evaluate both across different values of $\alpha \in [0,1]$, yielding ten additional baselines. Note that $\texttt{Greedy}(1) = \texttt{GreedyRemainder}(1)$ simply exhausts the entire budget in the first round. On real-world recruitment networks, our proposed method consistently outperforms these baselines on real-world recruitment networks (see the plot in https://github.com/icml2026authors/Rebuttal-for-Adaptive-Multi-Round-Allocation-with-Stochastic-Arrivals/blob/main/rebuttal.png), demonstrating the benefit of multi-round planning beyond myopic allocation.
>
> For intuition, **we proved that greedy is optimal for any given per-round budget**. If the optimal cross-round budget schedule were known, greedy under that schedule would be optimal. Thus, these baselines are strong comparators: they isolate the effect of budget scheduling while using optimal within-round decisions. Our gains therefore arise from a better state-dependent allocation schedule across rounds.
>
> # Functional Form (Q1)
>
> We interpret this question in two possible ways.
>
> (1) If the question concerns whether a parametric functional form (e.g., $f(\mathrm{covariates}) + \varepsilon$) simplifies the surrogate construction: the planning layer depends only on the induced distributional objects $\mathcal{D}_i$ (for the current frontier) and $\mathcal{P}$ (for future recruits). Thus, if such a model produces these distributions, the surrogate formulation remains unchanged. The main difficulty lies not in specifying these distributions, but in reasoning about the exact continuation value over future distribution-valued frontiers, which motivates the surrogate approximation in Section 6.
>
> (2) If the question concerns modeling dependencies (e.g., correlations between recruiters and recruits): we agree that real-world recruitment processes may exhibit such structure, but our modeling choice is a deliberate compromise between realism and tractability that isolates the key dynamic feature of the problem. Also, section 7 quantifies robustness to the extent such dependence appears as misspecification of the planning distribution. Please see our response to Reviewer F1GZ ("Our modeling choice") for a more detailed discussion.
>
> # References
>
> References [3,4,5,6,8] are given in our response to Reviewer umWA.

---

> > ### Author Rebuttal · Reviewer_127x · 2026-04-07
> >
> > Thank you for the clarifications. However, my original evaluation still stands

---

### Official Review · Reviewer_F1GZ · 2026-03-11

**Soundness:** 3
**Presentation:** 3
**Significance:** 2
**Originality:** 3
**Overall Recommendation:** 4
**Confidence:** 3

**Summary:**

This paper studies a resource allocation problem with a budget constraint and a random horizon. When the individual referral distributions and the population distribution are known, the authors derive tractable results for both the single-round and multi-round settings. When these distributions are unknown, the paper proposes an algorithm along with corresponding regret guarantees in terms of estimation error. Comprehensive numerical experiments are provided to validate the effectiveness of the proposed approach.

**Compliance With Llm Reviewing Policy:**

Affirmed.

**Key Questions For Authors:**

1. The authors may discuss more using examples about the population and individual referral distributions in practice, and explain the reason why the recruited number at the current round will determine the size of the frontier at the next round.

2. The decision problem has a random terminal time. Please provide the boundary (terminal time) condition for the Bellman equation.

3. A tighter and more clearly motivated characterization of the surrogate approximation error would strengthen the paper, as the current result appears closer to a conservative heterogeneity-based upper bound than to a sharp description of the aggregation loss. In addition, the constant in the frontier-estimation term of Theorem 7.2 appears inconsistent with the corresponding proof and would benefit from careful verification.

4. The titles of Sections 6.1 and 6.2 are the same.  There is a typo about $\hat D_{1:n}$ in the line above Equation (7).

**Limitations:**

yes

**Strengths And Weaknesses:**

Strengths

- This paper establishes a complete framework for a class of resource allocation problems. By utilizing the specific structure of the multi-round problem, this paper overcomes the possible error accumulations from Bellman recursion and provides an implementable computation method.

Weaknesses

- Although the authors introduce the 95-95-95 HIV target and explain the connection between the individual referral distributions and the distributions conditioned on covariates, I still find the motivations of the proposed problem are not very clear (see the questions below).
- The assumption of referral heterogeneity is far too strong and substantially departs from the real problem setting. Individual heterogeneity appears only within the current round, while the model assumes no temporal dependence between the referral distributions of currently recruited individuals and those in future rounds. This effectively removes the main source of dynamic structural complexity in referral networks, rather than providing a mild abstraction of it.

---

> ### Author Rebuttal · Authors · 2026-03-30
>
> We appreciate your time and effort in reviewing our work. We address your concerns below; please let us know if you require further clarifications. Thank you!
>
> # Real-world motivation and application (W1, Q1)
>
> The model is directly motivated by respondent-driven sampling (RDS) [1,2], a network-based chain-referral design for hard-to-reach populations in which a small set of initial participants ("seeds") is given a limited number of referral opportunities, and successful recruits become recruiters in the next wave. This mechanism is widely used in public-health studies [3,4,5,6]. For example, [6] uses RDS to recruit individuals with methamphetamine use disorder in China: seeds receive recruitment cards and incentives, can refer up to five peers, and recruitment proceeds over multiple rounds.
>
> Thus, unlike broadcast advertising, the set of individuals available in future rounds is *endogenously determined* by current recruitment outcomes. Network-based approaches such as RDS are recommended by WHO [7] because centralized outreach is often ineffective for hard-to-reach or stigmatized populations, whereas peer referral leverages social networks under tight resource constraints. This directly motivates our formulation, where allocating limited resources (e.g., coupons, incentives, or test kits) induces stochastic recruitment that determines future decision opportunities. We will revise the introduction to make this motivation more concrete.
>
> **We are also collaborating with domain-expert coauthors to evaluate such allocation strategies in a field study (details omitted for anonymity), further supporting the practical relevance of this setting.**
>
> # Our modeling choice (W2)
>
> We agree that real-world recruitment processes may exhibit richer temporal dependencies, such as correlations between recruiters and recruits. However, modeling such dependence would substantially enlarge the state space, make multi-round planning much harder, and require substantially richer data to estimate reliably, whereas in many recruitment settings the available data are limited.
>
> Our use of an **arbitrarily** heterogeneous frontier $\mathcal{D}_{1:n}$ and a population-level prior $\mathcal{P}$ is therefore intended to isolate the core dynamic feature of the problem, that current allocation decisions shape future decision opportunities, while still yielding a policy that can be analyzed and optimized in a principled way. This abstraction was developed in collaboration with domain-expert coauthors involved in field deployments, and should be viewed as a deliberate compromise between fidelity and tractability, rather than as a claim that such dependence is absent in practice.
>
> Experiment 6 also provides empirical evidence of robustness: the policy plans using a learned empirical population model but is executed directly on the unknown underlying graph, creating a mismatch between the planning model and true recruitment dynamics, yet our method still consistently outperforms the baselines.
>
> Our theoretical analysis does not require the planning model to be exactly correct. Section 7 explicitly studies misspecification in both the frontier and population distributions. To the extent that unmodeled temporal dependence induces a mismatch between the true future-arrival process and the planning distribution, Theorem 7.2 provides a way to quantify its impact. Stronger and more systematic dependence is an important direction for future work.
>
> # Boundary condition for Bellman (Q2)
>
> The process terminates when either the budget is exhausted or the frontier becomes empty (i.e., no new arrivals). Correspondingly, the Bellman recursion uses boundary conditions $V(0, \cdot) = 0$ and $V(\cdot, 0) = 0$, which is standard in budgeted dynamic programs [9,10].
>
> # Theorem 7.2 (Q3)
>
> The goal of Theorem 7.2 is to decompose performance loss into different sources of error, including estimation error and intrinsic surrogate error. Without additional structural assumptions on $\mathcal{D}$ and $\mathcal{P}$, it is difficult to obtain substantially tighter bounds, since our analysis is fully **distribution-agnostic**. Identifying tractable distribution families that admit sharper guarantees is an interesting direction for future work.
>
> Thank you for pointing out the inconsistency in Theorem 7.2. The proof is correct, and we will revise the theorem statement accordingly.
>
> # Typo and writing suggestions (Q4)
>
> Thank you for pointing these out. We will rename Section 6.1 to "Single round population-level surrogate" and fix the typo.
>
> # References
>
> References [1,2,7] are in our response to Reviewer SG33, and [3,4,5,6] in our response to Reviewer umWA.
>
> [9] Craig Boutilier, and Tyler Lu. Budget Allocation using Weakly Coupled, Constrained Markov Decision Processes. UAI, 2016.
>
> [10] Nicolas Carrara, Edouard Leurent, Romain Laroche, Tanguy Urvoy, Odalric-Ambrym Maillard, and Olivier Pietquin. Budgeted reinforcement learning in continuous state space. NeurIPS, 2019.

---

> > ### Author Rebuttal · Reviewer_F1GZ · 2026-04-03
> >
> > Thank you for the detailed clarifications. I will keep my scores unchanged.

---

> > > ### Author Response · Authors · 2026-04-06
> > >
> > > We sincerely appreciate your follow-up and are glad that our clarifications have adequately addressed your concerns! Thank you again for your positive recommendation and thoughtful feedback, and we hope you can recommend our work for acceptance.

---

### Official Review · Reviewer_SG33 · 2026-03-13

**Soundness:** 3
**Presentation:** 3
**Significance:** 2
**Originality:** 4
**Overall Recommendation:** 4
**Confidence:** 4

**Summary:**

This paper studies how to allocate resources across multiple rounds of recruitment. Its key modeling feature is that allocation decisions in the current round affect not only immediate rewards but also the pool of potential recruits in future rounds. The paper shows that the optimal single-round policy is greedy, develops a dynamic program based on population-level information for the multi-round setting, analyzes robustness to estimation error, and evaluates the approach in both synthetic and real recruitment environments.

**Compliance With Llm Reviewing Policy:**

Affirmed.

**Final Justification:**

My concerns had been adequately addressed and I raised my score.

**Key Questions For Authors:**

Questions:
1. I noticed what appears to be a typographical error: Sections 6.1 and 6.2 have exactly the same title. My understanding is that Section 6.1 is primarily concerned with the analysis of the optimal strategy for a single round under the population-level assumption, so the current duplication in titles may be misleading.
2. In Section 6, the original Bellman equation is reduced to a formulation that depends only on the current frontier size, rather than on the full distribution over individuals. It is not entirely clear whether this reduction is without loss of optimality, or whether it introduces an approximation.
3. The introduction could do more to motivate the model and make the setting accessible to readers. At present, concepts such as adaptive network-based recruitment are introduced rather abstractly, which makes it difficult to develop intuition for the problem. A clearer presentation grounded in a concrete application would better highlight both the distinctive features of the model and its practical relevance. It would also be helpful to explain more explicitly how the framework connects to machine learning and internet-based applications.
4. The relationship between Sections 5 and 6 could be presented more clearly. In their current form, the two sections appear to be organized in parallel, but substantively they play rather different roles in the paper. Reorganizing the exposition to better reflect their logical relationship would improve readability.
5. I would suggest renaming Section 7 as “Robustness Analysis”, which seems to better capture the content of that section.
6. The experimental comparison in Section 8 is currently limited to a small number of constant-allocation policies, which seems somewhat narrow. The empirical evaluation would be more convincing if it included additional rule-based benchmarks and, if feasible, learning-based baselines as well, such as neural-network-based policies.

**Strengths And Weaknesses:**

Strength:
The main strength of this paper is that it studies a genuinely novel and interesting dynamic decision problem: current allocation decisions affect not only immediate rewards but also the future pool of recruitable individuals. This intertemporal linkage makes the setting both practically relevant and theoretically nontrivial. In addition, the paper is rich and well-rounded in scope. It develops the analysis for both the single-round and multi-round settings, and further strengthens the overall contribution with robustness analysis and extensive simulation results in the final sections. Taken together, the paper offers a fairly complete treatment of the problem from modeling, theory, to empirical validation.

Weakness:
The main weakness of the paper is that, while the proposed model is certainly original, its economic interpretation remains somewhat unclear. In particular, it is difficult to see how the model maps cleanly to realistic applications.

---

> ### Author Rebuttal · Authors · 2026-03-30
>
> We appreciate your time and effort in reviewing our work. We address your concerns below; please let us know if you require further clarifications. Thank you!
>
> # Real-world motivation and application (Weakness, Q3)
>
> The model is directly motivated by respondent-driven sampling (RDS) [1,2], a network-based chain-referral design for hard-to-reach populations in which a small set of initial participants ("seeds") is given a limited number of referral opportunities, and successful recruits become recruiters in the next wave. This mechanism is widely used in public-health studies [3,4,5,6]. For example, [6] uses RDS to recruit individuals with methamphetamine use disorder in China: seeds receive recruitment cards and incentives, can refer up to five peers, and recruitment proceeds over multiple rounds.
>
> Thus, unlike broadcast advertising, the set of individuals available in future rounds is *endogenously determined* by current recruitment outcomes. Network-based approaches such as RDS are recommended by WHO [7] because centralized outreach is often ineffective for hard-to-reach or stigmatized populations, whereas peer referral leverages social networks under tight resource constraints. This directly motivates our formulation, where allocating limited resources (e.g., coupons, incentives, or test kits) induces stochastic recruitment that determines future decision opportunities. We will revise the introduction to make this motivation more concrete.
>
> **We are also collaborating with domain-expert coauthors to evaluate such allocation strategies in a field study (details omitted for anonymity), further supporting the practical relevance of this setting.**
>
> # Experimental concerns (Q6)
>
> In practice, RDS implementations almost universally impose a fixed number of coupons per participant (typically 2–5) [3,4,5,6], as recommended in WHO operational guidelines [8]. This corresponds directly to constant-allocation policies in our model. Thus, constant allocation is not a weak baseline, but the standard policy used in practice.
>
> That said, we agree with the reviewers that more baselines are useful. Building on Reviewer 127x's suggestion, we tested two additional classes of policies: a non-adaptive policy $\texttt{Greedy}(\alpha)$ and an adaptive policy $\texttt{GreedyRemainder}(\alpha)$
>
> (i) $\texttt{Greedy}(\alpha)$ applies single-round greedy allocation with fixed budget $\alpha b$ per round
>
> (ii) $\texttt{GreedyRemainder}(\alpha)$ allocates a fraction $\alpha$ of the remaining budget each round.
>
> We evaluate both across different values of $\alpha \in [0,1]$, yielding ten additional baselines. Note that $\texttt{Greedy}(1) = \texttt{GreedyRemainder}(1)$ simply exhausts the entire budget in the first round. On real-world recruitment networks, our proposed method consistently outperforms these baselines on real-world recruitment networks (see the plot in https://github.com/icml2026authors/Rebuttal-for-Adaptive-Multi-Round-Allocation-with-Stochastic-Arrivals/blob/main/rebuttal.png), demonstrating the benefit of multi-round planning beyond myopic allocation.
>
> For intuition, **we proved that greedy is optimal for any given per-round budget**. If the optimal cross-round budget schedule were known, greedy under that schedule would be optimal. Thus, these baselines are strong comparators: they isolate the effect of budget scheduling while using optimal within-round decisions. Our gains therefore arise from a better state-dependent allocation schedule across rounds.
>
> # Surrogate Bellman (Q2)
>
> The reduction from Eq. (3) to the surrogate formulation in Eq. (6) is **not** without loss. Eq. (6) replaces the true continuation value with a tractable population-level surrogate $U_P(r-s, N_s^g)$. The approximation error is characterized in Theorem 7.2, where $\Vert \mathcal{D} - \bar{\mathcal{D}} \Vert$ captures the loss from aggregating frontier composition into its size. For further discussion, see our response to Reviewer umWA ("When is the surrogate a good approximation").
>
> # Typo and writing suggestions (Q1,4,5)
>
> Thank you for these suggestions. We will clarify the relationship between Sections 5 and 6 by presenting Section 5 as the exact multi-round Bellman formulation and Section 6 as its tractable surrogate approximation. We will also rename Section 6.1 to "Single-round population-level surrogate" and Section 7 to "Robustness Analysis".
>
> # References
>
> [1] Heckathorn, D. D. Respondent-Driven Sampling: A New Approach to the Study of Hidden Populations. Social Problems, 44(2):174–199, 1997.
>
> [2] Goel, S. and Salganik, M. J. Respondent-driven sampling as Markov chain Monte Carlo. Statistics in Medicine, 28 (17):2202–2229, 2009.
>
> References [3,4,5,6,8] are given in our response to Reviewer umWA.
>
> [7] World Health Organization. WHO recommends social network-based HIV testing approaches for key populations as part of partner services package, 2019. Available at https://www.who.int/publications/i/item/WHO-CDS-HIV-19.32.

---

> > ### Author Rebuttal · Reviewer_SG33 · 2026-04-04
> >
> > Thank you for your detailed response. I appreciate the clarification and will raise my score.

---

> > > ### Author Response · Authors · 2026-04-06
> > >
> > > We sincerely appreciate your follow-up and your decision to raise the score; please remember to do before the review period is over! We are glad that our response has adequately addressed your concerns. Thank you again for your positive recommendation and thoughtful feedback, and we hope you can recommend our work for acceptance.

---

### Decision · Program_Chairs · 2026-04-30

**Decision:**

Accept (regular)

**Comment:**

The paper studies a sequential resource allocation problem in which, at each round, the decision-maker observes the remaining budget and the frontier, that is, the distributions of the number of potential recruits associated with each individual. These distributions are defined through the known features or covariates of the individuals. The decision-maker then chooses how much budget to allocate to each individual at the frontier. Whenever a new individual is recruited, their distribution over the number of potential recruits is drawn i.i.d. across individuals and rounds. The objective is to maximize the discounted number of recruits subject to the budget constraint.

The paper begins by solving the one-round problem, for which it is straightforward to show that the greedy allocation rule is optimal. This result then motivates the use of dynamic programming to address the multi-round setting. Since the latter is intractable, the authors propose a heuristic that depends only on the population, leading to a dynamic program for a surrogate population-level value function.

All reviewers acknowledged the novelty of the model. At the same time, they raised questions about its practical relevance, concerns that were partly addressed by the authors. One strong assumption is the lack of dependence across rounds in the recruitment-potential distributions. In practice, one might expect that an individual belonging to a given population category would tend to recruit others from a similar category. That said, relaxing these modeling assumptions appears genuinely difficult.

A second concern relates to the accuracy of the approximation induced by the surrogate value function. The impact of this approximation is upper-bounded in Theorem 7.2, but it seems difficult to assess quantitatively in practice, as it depends on the heterogeneity of the population.

Overall, the paper is sound and well written, provided the authors incorporate the clarifications requested by the reviewers. The main novelty of the paper lies in the model itself, rather than in the theoretical results, which are more limited.